# Quantifying energy use efficiency via entropy production: A case study from longleaf pine ecosystems

Susanne Wiesner[1], Christina L. Staudhammer[1], Paul C. Stoy[2], Lindsay R. Boring[3,4], Gregory Starr[1]

[1]Department of Biological Sciences, University of Alabama, Tuscaloosa, AL. 35487. USA.

[2]Department of Land Resources and Environmental Sciences, Montana State University, Bozeman, MT 59717, USA

[3]Jones Ecological Research Center, Newton, GA 39870, USA.

[4]Odum School of Ecology, University of Georgia, Athens, GA, 30602, USA

*Correspondence to*: Gregory Starr (gstarr@ua.edu)

**Abstract.** Ecosystems are open systems that exchange matter and energy with their environment. They differ in their efficiency in doing so as a result of their location on Earth, structure, and disturbance, including anthropogenic legacy. Entropy has been proposed to be an effective metric to describe these differences as it relates energy use efficiencies of ecosystems to their thermodynamic environment (i.e. temperature), but has rarely been studied to understand how ecosystems with different disturbance legacies respond when confronted with environmental variability. We studied three sites in a longleaf pine ecosystem with varying levels of anthropogenic legacy and plant functional diversity, all of which were exposed to extreme drought. We quantified radiative ($eff_{rad}$), metabolic and overall entropy changes – as well as changes in exported to imported entropy ($eff_{flux}$) in response to drought disturbance and environmental variability using 24 total years of eddy covariance data (8 years per site). We show that structural and functional characteristics contribute to differences in energy use efficiencies at the three study sites. Our results demonstrate that ecosystem function during drought is modulated by decreased absorbed solar energy and variation in the partitioning of energy and entropy exports owing to differences in site enhanced vegetation index and/or soil water content. Low $eff_{rad}$ and metabolic entropy, and slow adjustment of $eff_{flux}$ at the anthropogenically altered site prolonged its recovery from drought by approximately one year. In contrast, stands with greater plant functional diversity (i.e., the ones that included both C3 and C4 species) adjusted their entropy exports when faced with drought, which accelerated their recovery. Our study provides a path forward for using entropy to determine ecosystem function across different global ecosystems.

## 1 Introduction

Ecosystems utilize resources, such as solar radiation, nutrients and water, to maintain a state far from thermodynamic equilibrium (Amthor, 2010; Beer et al., 2009; Finzi et al., 2007; Thomas et al., 2016). Understanding ecosystem resource use efficiency is crucial, as anthropogenic and climate induced changes around the globe continue to alter ecosystem structure and function (Haddeland et al., 2014; Porter et al., 2012; Reinmann and Hutyra, 2016; Thom et al., 2017).

Ecosystems are open and dynamic systems that exchange matter and energy with their surroundings as described by the ecosystem energy balance:

$$R_n = R_{s,in} - R_{s,out} + R_{l,in} - R_{l,out} = LE + H + G + M \tag{1.1}$$

where $R_n$ is net radiation, $R_{s,in}$ and $R_{s,out}$ are incident and upwelling shortwave radiation, and $R_{l,in}$ and $R_{l,out}$ are incoming and upwelling longwave radiation, respectively. The terms LE, H and G represent energy exports through latent heat, sensible heat and ground heat fluxes, respectively and M is an energy storage term comprised of changes in biomass accumulation through metabolic processes (Holdaway et al., 2010). M is often neglected due to the assumption of a steady state over longer periods and because M is much smaller in magnitude compared to other fluxes. However, M imposes a control on energy fluxes, like $R_n$, LE and H, through changes in leaf area and reflective properties, as well as through active biotic control in response to changes in environmental variables (i.e., stomata opening and closing due to water availability (Hammerle et al., 2013).

From equation 1, ecosystem energy exchange is a function of its thermodynamic environment - the heat transfer of a system with its surroundings - which differs based on the different mechanisms by which heat is transported: conduction, convection, radiation. Complicating our understanding of ecosystem energy dynamics is the fact that more frequent fluctuations in environmental variables are expected as a result of global climate change, including extreme events like droughts, which will alter the resource efficiency of ecosystems across the globe and with it their resilience (Franklin et al., 2016; Woodward et al., 2010).

It is hypothesized that ecosystems aim to optimize their energy use and thus maximize their balance of entropy production and entropy exports to avoid thermodynamic equilibrium (Schneider and Kay, 1994; Schymanski et al., 2010). The magnitude of entropy production and entropy fluxes in ecosystems depend on thermodynamic gradients (i.e., thermal gradients, chemical gradients, etc.) between organisms and their surroundings (Kleidon, 2010). Ecosystems invest energy to build more complex structures (i.e., self-sustainability; Müller and Kroll, 2011; Virgo and Harvey, 2007), which can enhance their entropy export and therefore keep the ecosystem far from thermodynamic equilibrium (Odum, 1988; Schneider and Kay, 1994; Holdaway et al., 2010; Skene, 2015). For example, forest stands with more vertical structure were found to be more efficient in harvesting available light, which consequently increased their productivity (Bohn and Huth, 2017; Hardiman et al., 2011). Productive sites with greater leaf area can maintain higher LE fluxes, which increases their entropy export (Meysman and Bruers, 2010, Brunsell et al., 2011); LE fluxes also maintain lower ecosystem surface temperatures and thereby greater entropy production. On the contrary, large values of H caused by surface temperatures that are greater than air temperatures, result in lower entropy production (LeMone et al., 2007). This has been shown in deforested landscapes (Bonan, 2008; Khanna et al. 2017), as well as comparative studies of different vegetation types and in ecosystems with heterogeneity in their vegetation distribution (Holdaway et al., 2010; Brunsell et al., 2011; Kuricheva et al., 2017).

Here, we evaluate how efficiently ecosystems use energy by assessing ecosystem entropy production as well as by quantifying the ratios in entropy imports and exports ($eff_{flux}$ and dS/dt) in three study ecosystems that represent an edaphic and management gradient. We do so by measuring their structural complexity over an eight-year period via the enhanced vegetation index (EVI) and variation in annual understory biomass, and in relation to the energy and entropy partitioning of incoming energy from solar radiation. We build upon the techniques proposed by Holdaway et al. (2010), Brunsell et al. (2011), and Stoy et al. (2014), by calculating entropy production and entropy fluxes within longleaf pine (*Pinus palustris* Mill.) ecosystems. The sites differed in ecosystem structure (i.e., basal area, Table 1) and plant functional diversity due in part to differences in soil water holding capacity, as well as different levels of anthropogenic legacy. The sites experienced severe drought in the beginning of this

study, which we used to quantify entropy exchanges in response to the disturbance. First, we compare and contrast differences in ecosystem energy fluxes (i.e., $R_n$, LE, H, G and the net ecosystem exchange of carbon dioxide [NEE]) and entropy fluxes ($J_{LE}$, $J_H$, $J_G$, metabolic entropy [$S_m$] and radiative entropy production [$\sigma$]) in response to changes in structural and environmental variables (EVI, soil water content [SWC], vapor pressure deficit [VPD], and precipitation). Next, we quantify how entropy exports and entropy production at the different sites adjust to changes in incoming entropy when exposed to drought. We do so by estimating radiative efficiency ($eff_{rad}$), the ratio of entropy production to an empirical maximum entropy production (MEP), and ratios of daily imported and exported entropy fluxes ($eff_{flux}$), as well as through the overall change in entropy ($dS/dt$) at the sites. We hypothesize that: (1) the xeric site will have a higher entropy flux from $J_H$ and $J_G$, but lower $S_m$ due to its lower EVI and lower basal area, which will result in more variable $dS/dt$ compared to the other sites; (2) the mesic site will maintain higher $eff_{rad}$ due to its greater structural complexity (i.e., plant functional diversity and basal area) and thus greater absorptive capacity for solar radiation compared to the other sites; (3) the intermediate site will have lower $eff_{rad}$ and $eff_{flux}$ compared to the mesic and xeric sites, as a result of its lower plant functional diversity (i.e. low abundance of C4 species) and structural complexity, causing lower absorption of solar radiation and export of entropy through LE.

## 2 Materials and Methods

### 2.1 Site description

This study was conducted at the Joseph W. Jones Ecological Research Center in southwestern Georgia, USA (31.2201° N, 84.4792° W) from January 2009 to December 2016. The three sites are maintained by frequent low intensity fire on a two-year return interval and were last burned in 2015 (Starr et al., 2016). The climate is humid subtropical with a mean annual precipitation of 1310 mm (Kirkman et al., 2001). Mean temperature extremes range from 3 °C to 16 °C in winter and 22 °C to 33 °C in summer (NCDC, 2011).

The three sites differ based on soil moisture availability as a result of differences in soil drainage. The mesic site lies on somewhat poorly drained sandy loam over sandy clay loam and clay textured soils (Goebel et al., 1997; 2001). Soils at the intermediate site are well drained and have a depth to the argillic horizon of ~165 cm (Goebel et al., 1997). The xeric site lies on well-drained deep sandy soils with no argillic horizon (Goebel et al., 1997). All sites are situated within 10 km of each other and have average elevations of 165, 155, and 160 m for the mesic, intermediate, and xeric sites, respectively.

Ninety-five-year-old longleaf pine trees (*Pinus palustris* Mill.) dominate the overstory of all sites, and overall basal area ($B_A$) and diameter at breast height (DBH) varied by site (Table 1). The overstories of each site also contain a small proportion of oak trees; the xeric site has the highest proportion with 22 %, versus 8 % and 7.7 % at the mesic and intermediate sites, respectively. The understory at the mesic and xeric sites is largely covered with perennial C4 grass species, such as wiregrass (*Aristida beyrichiana* [Trin.]), whereas woody species dominate the intermediate site. Composition and abundance of other plant species varies by site (Kirkman et al., 2001; 2016). Soil perturbation at the intermediate site affected species richness, so that wiregrass is almost absent.

We acquired EVI for 2009 through 2016 for all three sites from the online data pool at lpdaac.usgs.gov via the NASA Land Processes Distributed Active Archive Center (LP DAAC) and the USGS Earth Resources Observation and Science Center

(EROS), using MODIS Aqua and Terra data products (MYD13Q1 and MOD13Q1; DAAC, 2008) to quantify changes in ecosystem structure from disturbance. EVI products for the sites were available on an eight-day basis and linearly interpolated to obtain daily estimates. We also acquired Palmer Drought Severity Indices (PDSI) for Southwest Georgia from the National Oceanic and Atmospheric Administration data archive for 2009 to 2016 to identify the months of drought disturbance (Dai et al., 2004).

Understory composition and biomass was estimated annually from 2009 through 2013. Thereafter, the collection frequency became biannual, so that 2014 and 2016 were missing in the data collection. Understory biomass was estimated using $0.75$ m$^2$ clip plots, which were randomly located by tossing a plot frame from pre-installed litter trap positions (n = 20 per site; see Wiesner et al. 2018). All live and dead vegetation, smaller than 1 m in height was clipped and analyzed in our laboratory. Vegetation was classified by plant life form (here, forbs, ferns, legumes, wiregrass, other grasses, and woody plants), and each sample was dried to constant weight.

*Net ecosystem exchange of CO$_2$ measurements.* Net ecosystem exchange (NEE) was measured continuously at 10 Hz at all three sites from January 2009 to December 2016 using open-path eddy covariance (EC) techniques (Whelan et al., 2013). Data were stored on CR-5000 dataloggers (Campbell Scientific, Logan, UT). CO$_2$ and water vapor concentration were measured with an open path infrared gas analyzer (IRGA, LI-7500, LI-COR Inc., Lincoln, NE) and wind velocity and sonic temperature were measured with a three-dimensional sonic anemometer (CSAT3, Campbell Scientific, Logan, UT). These sensors were installed ~4 m above mean canopy height at each site (34.5, 37.5, and 34.9 m for the mesic, intermediate and xeric sites, respectively), ~0.2 m apart to minimize flow distortion between the two instruments and vertically aligned to match the sampling volume of both instruments.

## 2.2 Sensible and latent heat flux measurements

Net energy fluxes of LE and H were estimated in W m$^{-2}$ using temperature and wind velocity measurements from the sonic anemometer, as well as water vapor density measurements from the IRGA:

$$LE = \lambda \rho_a \overline{w'q'} \tag{2.1}$$

$$H = \rho_a c_p \left( \overline{w'T_s'} - 0.000321 T_s \overline{w'q'} \right) \tag{2.2}$$

where $\lambda$ is the latent heat of vaporization (J kg$^{-1}$), $\rho_a$ is the density of air (kg m$^{-3}$), $c_p$ is the specific heat of air (kJ kg$_{-1}$ K$^{-1}$), $w'$ is the instantaneous deviation of vertical wind speed (w, m s$^{-1}$) from the mean, and $q'$ and $T_s'$ are the instantaneous deviations of water vapor concentration (kg kg$^{-1}$) and sonic temperature (Kaimal and Gaynor, 1991) from their respective means. The overbars in Eqs. 2.1 and 2.2 signify the time-averaged covariance. Missing H and LE were gap-filled on a monthly basis using simple linear models as a function of $R_n$.

In cases where energy balance closure was not achieved, energy fluxes of H and LE were corrected using the Bowen method following Twine et al. (2000), where fluxes are adjusted using residual energy, and the estimated Bowen ratio ($\beta$ = H/LE), which assumes that $\beta$ was correctly measured by the EC system:

$$LE = \frac{1}{1+\beta} (R_n - G) \tag{2.3}$$

$$H = \beta \times LE \tag{2.4}$$

Closing the energy balance is important to quantify differences in energy and entropy fluxes by site, as according to the First law of Thermodynamics energy is always conserved. To quantify differences in environmental drivers and site variation between energy and entropy fluxes, we established models of average daily energy fluxes (described in section 2.7)

## 2.3 Meteorological instrumentation

Meteorological data above the canopy were also collected and stored on the CR-5000 dataloggers (Campbell Scientific, Logan, UT). Meteorological data measured on the towers included: photosynthetically active radiation (PAR; LI-190, LI-COR Inc., Lincoln, NE), global radiation (LI-200SZ, LI-COR Inc., Lincoln, NE), incident and outgoing shortwave and longwave radiation to calculate $R_n$ (NR01, Hukseflux, thermal sensors, Delft, The Netherlands), precipitation (TE525 Tipping Bucket Rain Gauge, Texas Electronics, Dallas, TX), wind direction and velocity (Model 05103-5, R.M. Young, Traverse City, MI), air temperature ($T_{air}$) and relative humidity (RH; HMP45C, Campbell Scientific, Logan, UT), and barometric pressure (PTB110, Vaisala, Helsinki, Finland).

Soil temperature ($T_{soil}$), volumetric water content of the soil (SWC) and soil heat flux (G) were measured in one location near the base of each tower at each site every 15 seconds and averaged every 30 minutes on an independently powered CR10X datalogger. $T_{soil}$ was measured at depths of 4 and 8 cm with insulated thermocouples (Type-T, Omega Engineering, INC., Stamford, CT), and G was measured at a depth of 10 cm with soil heat flux plates (HFP01, Hukesflux, Delft, The Netherlands). SWC was measured within the top 20 cm of the soil surface using a water content reflectometer probe (CS616, Campbell Scientific, Logan, UT).

## 2.4 Data processing

Raw EC data were processed using EdiRe (v.1.4.3.1184; Clement, 1999), which carried out a two-dimensional coordinate rotation of the horizontal wind velocities to obtain turbulence statistics perpendicular to the local streamline. Fluxes were calculated for half-hour intervals and then corrected for mass transfer resulting from changes in density not accounted for by the IRGA. Barometric pressure data were used to correct fluxes to standard atmospheric pressure. Flux data screening was applied to eliminate 30-min fluxes of NEE, H and LE, resulting from systematic errors as described in Whelan et al. (2013) and Starr et al. (2016). Such errors encompassed (amongst other things): rain, poor coupling of the canopy and the atmosphere (defined by the friction velocity, ustar), and excessive variation from half-hourly means.

Gross ecosystem exchange (GEE) and ecosystem respiration ($R_{eco}$) were estimated from eddy covariance measurements of net ecosystem exchange of $CO_2$ (NEE; $\mu$mol m$^{-2}$ s$^{-1}$) at a time resolution of half an hour, from which GEE and $R_{eco}$ can be estimated as follows:

$$GEE = -NEE + R_{eco} \tag{2.5}$$

Missing half hourly data were gap-filled as described in Whelan et al. (2013) and Starr et al. (2016). Daytime and nighttime data were estimated utilizing a Michaelis-Menten approach for (PAR > 10 $\mu$mol m$^{-2}$ s$^{-1}$) and a modification of the Lloyd and Taylor (1994) model (PAR $\leq$ 10 $\mu$mol m$^{-2}$ s$^{-1}$), respectively. Monthly equations were used to gap-fill data; however, where too

few observations were available to produce stable and biologically reasonable parameter estimates, annual equations were used. NEE partitioning to estimate daytime $R_{eco}$ was performed by using the nighttime gap-filling equation, and then utilizing equation (2.5) to estimate GEE. Nighttime GEE was assumed to be zero.

## 2.5 Entropy production calculations

5 Half-hourly GEE and $R_{eco}$ were converted to W m$^{-2}$ (GEE$_e$ and $R_{eco_e}$), using the assumption that one micromole of $CO_2$ stores approximately 0.506 J, where 1 J m$^{-2}$ s$^{-1}$ equals 1 W m$^{-2}$ (Nikolov et al., 1995), which is then released during respiration. For entropy production and fluxes of shortwave ($R_s$) and longwave radiation ($R_l$) we followed established approaches of Brunsell et al. (2011), Holdaway et al. (2010), and Stoy et al. (2014). The half-hourly entropy flux produced through absorption of $R_s$ emitted by the surface of the sun ($J_{Rs}$, W m$^{-2}$ K$^{-1}$) was calculated as:

$$J_{Rs} = \frac{R_{s,net}}{T_{sun}} \tag{2.6}$$

where sun surface temperature ($T_{sun}$) was assumed to be 5780 K, with $R_{s,net}$ defined as the difference of incident and upwelling $R_s$. The entropy flux of $R_l$ ($J_{Rl}$, W m$^{-2}$ K$^{-1}$) was calculated as:

$$J_{Rl} = \left( \frac{R_{l,in}}{T_{sky}} - \frac{R_{l,out}}{T_{srf}} \right) \tag{2.7}$$

where $R_{l,in}/T_{sky}$ is the entropy flux of $R_{l,in}$ as incoming $R_l$ ($J_{Rl,in}$), and $R_{l,out}/T_{srf}$ is the entropy flux of $R_{l,out}$ as outgoing $R_l$ ($J_{Rl,out}$).

15 Surface temperature ($T_{srf}$; K) was calculated from upwelling $R_l$ ($R_{l,out}$):

$$T_{srf} = \left( R_{l,out} \middle/ (A \times e_{srf} \times k_B) \right)^{1/4} \tag{2.8}$$

with emissivity of the surface calculated as $e_{srf} = 0.99 - 0.16\alpha$ (Juang et al., 2007), the view factor A was assumed to be unity, and the Stefan-Boltzmann constant $k_B = 5.67 \times 10^{-8}$ W m$^{-2}$ K$^{-4}$. The shortwave albedo ($\alpha$) was calculated as the daily average of noontime outgoing $R_s$ ($R_{s,out}$) divided by $R_{s,in}$. The sky temperature, $T_{sky}$ (K), was calculated from $R_{L,in}$ using the Stefan-

20 Boltzmann equation:

$$T_{sky} = \left( R_{l,in} \middle/ (A \times e_{atm} \times k_B) \right)^{1/4} \tag{2.9}$$

where the emissivity of the atmosphere ($e_{atm}$) was assumed to be 0.85, following Campbell and Norman (1998). All other ecosystem entropy fluxes $J_{LE}$, $J_H$, $J_G$, and $J_{GEE}$ and $J_{Reco}$ (W m$^{-2}$ K$^{-1}$) were calculated by dividing the energy fluxes by temperature as:

$$J_x = \frac{x}{T_y} \tag{2.10}$$

where x = LE, H, G and GEE$_e$ and $R_{eco_e}$, and $T_y$ = was assumed to be $T_{air}$ (for $J_{LE}$, $J_H$, $J_{GEE}$ and $J_{Reco}$; K) or $T_{soil}$ (for $J_G$, in K). We also calculated entropy produced from evaporation associated with mixing of saturated air from the canopy with the fraction of air in the atmosphere that has RH below 100 % ($JLE_{mix}$), following Holdaway et al. (2010):

$$JLE_{mix} = ET \times R_v \times \ln(RH) \tag{2.11}$$

where the evapotranspiration rate is calculated as $ET = LE/\lambda$ (kg m$^{-2}$ s$^{-1}$) and $R_v$ is the gas constant of water vapor (0.461 kJ kg$^{-1}$ K$^{-1}$ for moist air).

The sum of entropy of ecosystem fluxes ($J$, W m$^{-2}$ K$^{-1}$) for each half-hour was then calculated by adding all entropy fluxes between the surface and atmosphere:

$$J = J_{Rl} + J_{Rs} + J_{LE} + J_H + J_G + J_{GEE} + J_{Reco} + JLE_{mix} \tag{2.12}$$

The conversion of low entropy $R_s$ and $R_l$ to high entropy heat at the surface through absorption of $R_s$ and $R_l$, respectively, was calculated as:

$$\sigma_{Rs} = R_{s,net}\left(\frac{1}{T_{srf}} - \frac{1}{T_{sun}}\right) \tag{2.13}$$

$$\sigma_{Rl} = R_{l,in}\left(\frac{1}{T_{srf}} - \frac{1}{T_{sky}}\right) \tag{2.14}$$

where $T_{srf}$ is the radiometric surface temperature (Eq. 2.8) and $\sigma_{RS}$ and $\sigma_{Rl}$ are in W m$^{-2}$ K$^{-1}$.

The overall half-hourly entropy production ($\sigma$, W m$^{-2}$ K$^{-1}$) was then calculated as the sum of the entropy productions of $R_s$ and $R_l$:

$$\sigma = \sigma_{Rl} + \sigma_{Rs} \tag{2.15}$$

We excluded the factor 4/3, which is associated with the transfer of momentum exerted by electromagnetic radiation on a

surface (Wu et al., 2008), in our calculations of $\sigma$ and $J$ for entropy production and entropy fluxes because we assumed that radiation pressure at the sites would be negligible (see Ozawa et al. 2003; Kleidon and Lorenz, 2005; Fraedrich and Lunkeit, 2008; Kleidon, 2009; Pascale et al., 2012).

To account for the difference in absorbed radiation on leaf and non-vegetated surfaces, we partitioned $\sigma$ using EVI as an approximation for fractional vegetation cover. Accordingly, $\sigma$ of non-vegetated surfaces ($\sigma_{land}$) was estimated as:

$$\sigma_{land} = (1 - EVI) \times \sigma \tag{2.16}$$

Entropy production on leaf surfaces ($\sigma_{leaf}$, eq. 2.17) was calculated as the sum of entropy production ($\sigma_{PAR}$ eq. 2.18) from absorbed photosynthetic active radiation (FPAR in W m$^{-2}$, eq. 2.19), and entropy production from the remainder of $R_s$-PAR ($\sigma_{Rs,leaf}$, eq. 2.20), assuming all was absorbed and converted into heat on leaf surfaces, as well as entropy production from absorbed longwave radiation on leaf surfaces (eq. 2.21).

$$\sigma_{leaf} = \sigma_{PAR} + \sigma_{Rs,leaf} + \sigma_{Rl,leaf} \tag{2.17}$$

where,

$$\sigma_{PAR} = FPAR\left(\frac{1}{T_{air}} - \frac{1}{T_{sun}}\right) \tag{2.18}$$

$$FPAR = EVI \times PAR \tag{2.19}$$

$$\sigma_{Rs,leaf} = (R_s - PAR)\left(\frac{1}{T_{air}} - \frac{1}{T_{sun}}\right) \times EVI \tag{2.20}$$

$$\sigma_{Rl,leaf} = \sigma_{Rl} \times EVI \tag{2.21}.$$

We assumed $T_{air}$ was close to leaf temperature. While this formulation may introduce assumptions about the absorptive behavior of leaves, it helps us to estimate entropy production from the metabolic processes of photosynthesis and respiration ($S_m$) as follows:

$$S_m = \sigma_{leaf} + J_{GEE} + J_{Reco} \tag{2.22}$$

Finally, we estimated half-hourly change in entropy production (S) over time (t) in W m$^{-2}$ K$^{-1}$ of the ecosystem by adding entropy flux of imports ($J_{Rs,net}$, $R_{Rl,in}$) and exports (i.e., $J_{LE}$, $J_H$, $J_G$, $J_{GEE}$, $J_{Reco}$, $J_{Rl,up}$, $JLE_{mix}$) and entropy production of vegetated and non-vegetated surfaces:

$$dS/dt = J + \sigma_{land} + \sigma_{leaf} \tag{2.23}$$

Note that this approach does not account for entropy production due to frictional dissipation of entropy from rainfall or subsurface water flow, as these would be of even smaller magnitude than entropy production from metabolic activity of the ecosystem (Brunsell et al., 2011). Here negative dS/dt represents the export of entropy of the ecosystem to its surroundings.

## 2.6 Ecosystem entropy models for radiation and ecosystem fluxes

We estimated half-hourly MEP of the radiation budget ($MEP_{rad}$) in W m$^{-2}$ K$^{-1}$, to compare site differences in radiation energy use and entropy dissipation.

Empirical MEP ($MEP_{rad}$) was determined following Stoy et al. (2014), by estimating the MEP of half-hourly $R_s$ ($MEP_{Rs}$) and $R_l$ ($MEP_{Rl}$):

$$MEP_{Rs} = R_{s,in} \left( \frac{1}{T_{srf}} - \frac{1}{T_{sun}} \right) \tag{2.23}$$

$$MEP_{Rl} = R_{l,net} \left( \frac{1}{T_{srf}} - \frac{1}{T_{air}} \right) \tag{2.24}$$

$$MEP_{rad} = MEP_{Rs} + MEP_{Rl} \tag{2.25}$$

This method offers a means to compare different sites with respect to their reflective and absorptive capacities *versus* a reference ecosystem that absorbs and dissipates all incident solar energy. Note that $MEP_{Rl}$ is often of lower magnitude than $MEP_{Rs}$ because here we assume that an efficient ecosystem would dissipate less energy through sensible heat, such that $T_{srf}$ would approach $T_{air}$.

The half-hourly entropy ratio of radiation is then calculated using $\sigma_{land}$ and $\sigma_{leaf}$ as follows:

$$eff_{rad} = \frac{\sigma_{land} + \sigma_{leaf}}{MEP_{rad}} \tag{2.26}.$$

We refer to this ratio as an efficiency to describe differences in the absorptive characteristics at the sites, where a ratio closer to 1 would indicate high radiation absorption. Furthermore, sites that maintain lower surface temperatures through greater LE fluxes would also increase their entropy production, thus linking ecosystem functional efficiency with radiative entropy production. We then estimated the variable $eff_{flux}$ as the ratio of incoming radiation entropy ($J_{Rs}$ and $J_{Rl,in}$) and the sum of exported entropy fluxes ($J_{LE}$, $J_H$, $J_G$, $J_{GEE}$, $J_{Reco}$, and $J_{Rl,up}$) to assess how entropy was partitioned into entropy production and entropy fluxes over the different study years.

## 2.7 Statistical analyses

We estimated average daily values for all response variables to decrease autocorrelation for statistical analysis. We first tested for significant differences in environmental and structural variables among the three sites prior to the entropy analysis. We estimated simple general linear mixed models (GLMM) using the R package *nmle* to look at differences among sites for: rain, SWC, vapor pressure deficit (VPD), EVI, $T_{srf}$, $T_{air}$, $T_{sky}$ and $T_{soil}$, as well as $R_{s,in}$, $R_{s,out}$, $R_{l,in}$ and $R_{l,out}$. All response variables were daily means. For rainfall we calculated monthly sums to estimate differences among the sites. We included a random effect for day of measurement, to account for repeated measurements, as well as an AR(1) structure to account for temporal autocorrelation among measurements. The model of rainfall only included year and site as independent variables and no random effects. Independent variables for the other models were month, year and site, as well as their interactions.

Subsequently, we estimated GLMMs of daily energy ($R_n$, LE, H, G and $NEE_e$) and entropy fluxes ($J_{LE}$, $J_H$, $J_G$, and $S_m$), entropy production ($\sigma$), entropy ratios ($eff_{rad}$ and $eff_{flux}$) and overall entropy (dS/dt) to quantify their differences by environmental and structural variables by site. For all models we included random effects and an AR(1) autoregressive correlation structure to account for repeated daily measurements. All models initially included independent variables for site, year and month, mean EVI, SWC, VPD and daily rainfall sums. We also included interactions of environmental variables with site, site with year and site with month, to determine changes in the energy efficiency over the study period among sites. Independent variables and their interactions were deemed significant when p<0.05. We used a Tukey adjustment to test for significant differences among sites. GLMM analyses were performed via the R packages *nlme*, *lsmeans*, and *car* (Fox and Weisberg, 2011; Lenth, 2016; Pinheiro et al., 2014).

## 3 Results

### 3.1 Differences in environmental, radiative and temperature variables among sites

All three sites experienced a severe drought from mid-2010 through mid-2012 (Fig. S1, Supplementary Information). There was no significant difference between the mesic and xeric sites in rainfall sums, but the intermediate site had lower rainfall sums (~20 mm per month) compared to the other sites (Table S1). SWC was significantly lower at the xeric (<19 %) compared to mesic and intermediate sites (~20 %) for all years of this study (Fig. 1a and b, Table S2). SWC and EVI decreased during the drought at all sites, but only significantly so at the mesic site. VPD significantly increased at all sites during the drought. For all years, EVI was significantly lower (0.02-0.04) at the xeric site compared to the other two sites (Fig. 1e and f), while the intermediate site had significantly higher EVI compared to the mesic site, except in 2010.

Daily $T_{srf}$ at the mesic site was significantly higher than the xeric site for all years except 2012, 2014 and 2016 (Fig. 2a). From 2012 to 2016 the intermediate site had higher $T_{srf}$ compared to the other two sites. $T_{air}$ was significantly lower at the mesic site compared to the intermediate and xeric sites for all years, except in 2012, and in 2014, when the xeric site had higher $T_{air}$ compared to the intermediate (Fig. 2a). $T_{soil}$ was significantly lower at the mesic site compared to the other sites, except in 2013, when there was no significant difference between the mesic and xeric sites. For all years, daily $T_{soil}$ was significantly

higher at the xeric site compared to the intermediate site except for 2011 and 2012, when the intermediate site was significantly higher.

$R_{s,out}$ was significantly higher at the xeric site compared to the other sites, except for 2014, where we found no significant difference between the intermediate and xeric sites. Daily $R_{s,out}$ was also significantly lower at the mesic site, compared to the

intermediate site, except in 2009. Average daily $R_{l,out}$ was significantly lower at the mesic site compared to the intermediate site during all years, except for 2011 and 2012, and compared to the xeric site for all years, except for 2011. The intermediate site had significantly higher $R_{l,out}$ compared to the xeric site during 2013, 2014 and 2016. As a consequence of these component fluxes, $R_n$ was significantly higher at the xeric site compared to the intermediate site for all years except 2009 and 2014 (SI Fig. S2a, Table S3). Average $R_n$ was significantly lower at the mesic site compared to the xeric site in 2013 and 2016, and was

significantly higher compared to the xeric site from 2009 to 2011.  Average daily $R_n$ significantly increased at the intermediate and xeric sites but showed no change at the mesic site with an increase in EVI (SI Fig. S3a).

Environmental, radiative and temperature variables also tended to be significantly different among months within site, and in many instances among sites by month. Differences followed seasonal patterns, as noted in SI Fig. S2 and SI Table S2.

### 3.2 Understory wiregrass and woody abundance at the sites

Wiregrass was virtually absent at the intermediate site for all years of this study (Fig. 4a), whereas woody species were more abundant compared to the others. The mesic and xeric sites both had higher proportions of wiregrass in the understory (~25 % versus 5 % at the intermediate site), which slightly decreased during 2011 (Fig. 4a). In addition, woody biomass increased to ~75 g m$^{-2}$ at the xeric site during 2011, but not at the mesic site. In 2012, woody biomass decreased to ~40 g m$^{-2}$ at the xeric and intermediate sites and remained low during the following years at the xeric site, but increased at the intermediate site (>100

20   g m$^{-2}$, Fig. 4b).

### 3.3 Energy fluxes of H, LE, and G

LE was significantly lower at the intermediate site compared to the mesic site for all years, except 2011, and compared to the xeric site for all years, except for 2015. We found no significant difference between the mesic and xeric sites in 2009, 2010, 2014 and 2016, but for the other years of this study the xeric site had significantly higher LE. LE significantly increased at all

sites with higher EVI, with a greater increase at the intermediate and a smaller increase at the xeric site, compared to the mesic site (SI Fig. S3g). LE significantly increased at all sites with an increase in SWC and VPD (SI Fig. S3e and f). LE at the intermediate site was significantly lower compared to the other sites for all levels of VPD (SI Fig. S3g). LE was significantly lower with higher rainfall, with no significant differences among sites (SI Fig. S3h).

There was no significant difference in H between the mesic and intermediate sites, except in 2011 and 2013, when the mesic

site was higher than the intermediate, and in 2015 and 2016, when the reverse occurred. H was significantly lower at the xeric site compared to the mesic site for all years except for 2014 and 2016, and compared to the intermediate site for all years except 2011 and 2013. Average H was significantly higher at the mesic site compared to the xeric site during the months of May through October (SI Fig. S2b). The intermediate site had significantly lower H compared to the other two sites for the

months of January through March and the xeric site had significantly lower H for June through October. Compared to the other two sites, average H was significantly lower at the intermediate site when EVI was greater than 0.4, and significantly higher at the xeric site for EVI > 0.5 (SI Fig. S3i). Average H significantly decreased at all sites with an increase in SWC (SI Fig. S3j). Average daily H significantly increased at all sites with an increase in VPD, with a lower decrease at the intermediate site (SI Fig. S3k).

G was significantly lower at the intermediate site during 2016 (negative), compared to 2009 through 2011 and 2014. Average daily G was positive during summer months, and negative during winter months (October through March) at all sites (SI Fig. S2b). Average daily G significantly decreased with an increase in EVI at the mesic and intermediate site, but had no significant change at the xeric site (SI Fig. S3m). G was significantly less positive at the xeric site compared to the other sites for EVI < 0.3, but was significantly more negative at the intermediate site compared to the mesic and xeric sites when EVI was above 0.4. Average G significantly decreased (to negative) with an increase in SWC (SI Fig. S3n), and significantly increased (to positive) with an increase in VPD, but only at the intermediate and xeric sites (SI Fig. S3o). Daily rainfall did not significantly alter G at the sites, but the intermediate site had significantly more negative G compared to the other two sites (2-10 W m$^{-2}$) when daily rainfall was positive (SI Fig. S3p).

### 3.4 Entropy production and fluxes of $J_H$, $J_{LE}$, and $J_G$

For all years, average daily σ (as the sum of $σ_{land}$ and $σ_{leaf}$) was significantly higher at the mesic site compared to the intermediate site (by > 0.01 – 0.036 W m$^{-2}$ K$^{-1}$; Fig. 5a, Table S4), while σ was not significantly different between the mesic and xeric sites for almost all years (Fig 5a). Average daily σ significantly increased with EVI, independent of site (Fig. 6a), and also significantly increased with SWC and VPD, with a greater slope at the xeric site (Fig. 6b and c). Average daily σ significantly decreased at all sites with an increase in rainfall (noting that entropy production from rainfall itself is not considered here and assumed to be approximately equal among ecosystems), and σ was significantly lower at the intermediate site during rainy periods compared to the other two sites (Fig. 6d). There was no significant difference in σ at the mesic and xeric sites for all levels of rain.

The xeric site had significantly higher average daily $J_{LE}$, ranging from ~0.22 to 0.28 W m$^{-2}$ K$^{-1}$, versus the intermediate site with ~0.18 – 0.25 W m$^{-2}$ K$^{-1}$ (Fig. 5a, Table S4) for all years, except 2015. $J_{LE}$ at the xeric site was also higher than the mesic site in 2011 through 2013 and in 2015, ranging from 0.2 to 0.26 W m$^{-2}$ K$^{-1}$. The mesic site had ~0.01-0.06 W m$^{-2}$ K$^{-1}$ higher $J_{LE}$ compared to the intermediate site, except in 2011. $J_{LE}$ significantly increased with greater EVI and SWC (Fig. 6e and f). $J_{LE}$ was significantly higher at the xeric site compared to the other sites for EVI < 0.4. $J_{LE}$ was significantly higher at the xeric site compared to the other sites when SWC was above 19%, similar to the model of LE. $J_{LE}$ significantly increased with VPD, and significantly decreased with rainfall (Fig. 6g and h). Unlike the model results for LE, the effects of VPD were not significantly different by site.

Models of H and $J_H$ were similar, except that $J_H$ in the mesic and xeric sites were not significantly different in 2015 (Fig. 5a, Table S4). Average daily $J_H$ was significantly higher at the mesic site in 2011 and 2012 (~0.2-0.24 W m$^{-2}$ K$^{-1}$) compared to the intermediate (~0.19 W m$^{-2}$ K$^{-1}$; Fig. 5a) and xeric sites (~0.16-0.20 W m$^{-2}$ K$^{-1}$). In 2009, 2010 and 2012, the xeric site had

significantly lower $J_H$ compared to the other sites (by $\sim 0.02$ W m$^{-2}$ K$^{-1}$). $J_H$ decreased only at the mesic and intermediate sites with increasing EVI (Fig. 6i) such that the intermediate site had significantly lower $J_H$ compared to the other sites when EVI was above 0.4. $J_H$ decreased with increased SWC at all sites, and the xeric site had significantly lower $J_H$ compared to the other sites when SWC was above 19 % (Fig. 6j). VPD significantly increased $J_H$ at all three sites, with a greater increase at the xeric site (Fig. 6k). $J_H$ significantly decreased at all sites with increased rainfall, where the intermediate site had significantly lower $J_H$ compared to the mesic and xeric sites when rainfall was greater than 40 mm per day (Fig. 6l).

Average daily $J_G$ was not significantly different among the years 2009-2014 and 2016 at the mesic site, but significantly increased during 2015 (Fig. 5a, Table S4), similar to the model results for G. Similarly, $J_G$ was significantly lower at the intermediate site during 2016 (negative). $J_G$ at the xeric site was not significantly different by year. Average daily $J_G$ was positive during summer months, and negative during winter months at all sites (Fig. 5b). Average daily $J_G$ significantly decreased from positive to negative at the mesic and intermediate sites with an increase in EVI, with no significant change at the xeric site (Fig. 6m), similar to the model of G. $J_G$ was significantly more negative at the intermediate site compared to the other sites for EVI > 0.4. Average $J_G$ only significantly decreased at the intermediate and xeric sites (to negative), such that $J_G$ was significantly more negative at the two sites when SWC was above 18% (Fig. 6n). $J_G$ significantly increased with greater VPD, independent of site (Fig. 6o). Similar to the model of G, daily rainfall did not significantly alter the magnitude of $J_G$ at the sites. However, the intermediate had significantly more negative $J_G$ compared to the other two sites when daily rainfall increased (Fig. 6p).

**3.5 Metabolic energy and entropy**

Metabolic energy was consistently more negative (more energy uptake) at the mesic site, compared to the other sites for all years in this study (Fig 7a, Table S5). The intermediate and xeric sites exported metabolic energy from 2009 through 2011, which was greater at the intermediate site for 2010. $NEE_e$ significantly increased to more negative at all sites during May and significantly decreased during August through October, which resulted in positive $NEE_e$ at the intermediate site (Fig. 7b). $NEE_e$ significantly decreased at all sites with an increase in EVI, which was greater at the xeric site (Fig. 7c). An increase in SWC resulted in decreasing $NEE_e$, independent of site (Fig. 7d). An increase in VPD significantly decreased $NEE_e$ to more negative at all sites, with a greater decrease at the intermediate site (Fig. 7e). Increases in rainfall significantly increased $NEE_e$ to positive at all sites, where the intermediate site had a greater increase compared to the other sites (Fig. 7f).

Results of the model of $S_m$ indicated that the mesic site had significantly greater metabolic entropy production compared to the intermediate site for all years but 2009 and 2013. The xeric site had significantly greater $S_m$ compared to the mesic site in 2012 through 2014 and in 2016, and compared to the intermediate site for all years (Fig. 7g). $S_m$ was greater during summer months at all sites with no significant differences between the mesic and xeric sites from February through August, but significantly lower at the intermediate site compared to the xeric site for all months (Fig. 7h, Table S5). Metabolic entropy production was significantly lower at the intermediate site compared to the mesic site for most months except January, April, October and December. Values of $S_m$ significantly increased with an increase in EVI, independent of site (Fig. 7i). SWC significantly increased $S_m$ at all sites, with a greater slope at the xeric site (Fig. 7j). Higher VPD significantly increased $S_m$

similar to the model of $NEE_e$; however slopes were more similar among the sites (Fig. 7k). Rainfall significantly decreased $S_m$ to ~0 with a greater slope at the intermediate site, similar to the model of $NEE_e$ (Fig. 7l).

## 3.6 Entropy models

From 2011 through 2016, $eff_{rad}$ was significantly higher at the mesic site (0.89-0.93), compared to the intermediate (0.88-0.91) and xeric (0.88-0.92) sites, which were not significantly different (Fig. 8a). Average $eff_{rad}$ did not significantly change with EVI or SWC. Higher VPD significantly decreased values of $eff_{rad}$ at all sites (Fig. 8c). The mesic site had significantly higher values of $eff_{rad}$ compared to the other two sites for all levels of VPD (Fig. 8c). Rainfall significantly increased values of $eff_{rad}$ at all sites, with a greater increase at the intermediate site (Fig. 8d, Table S6).

Daily average $eff_{flux}$ was significantly greater at the mesic site for most of the measurement period (Fig. 9a, Table S6). $eff_{flux}$ was significantly higher at the xeric site compared to the intermediate site for the years 2009, 2011, and 2013 through 2015. For 2012 and 2016 the intermediate site had significantly greater $eff_{flux}$ compared to the xeric site. Greater EVI only significantly increased $eff_{flux}$ at the mesic site, which had higher $eff_{flux}$ compared to the other sites for all levels of EVI (Fig. 9c). The intermediate site had significantly lower $eff_{flux}$ compared to the xeric site when EVI was above 0.3. An increase in SWC significantly decreased values of $eff_{flux}$ only at the intermediate and xeric sites, with a greater decrease at the xeric site (Fig. 9d). Higher VPD significantly decreased $eff_{flux}$ at all sites, with a greater decrease at the intermediate site (Fig. 9e). Rainfall significantly increased $eff_{flux}$ at all sites, where the intermediate site showed the highest increase (Fig. 9f).

There was no significant difference in dS/dt among sites for all years and months, except in 2014, where the intermediate site had significantly higher dS/dt compared to the other sites (Fig. 10a, Table S6). In addition, the xeric site accumulated dS/dt during 2012 such that it was significantly different from the other sites. An increase in VPD resulted in a significant increase in dS/dt (more entropy export), independent of site (Fig. 10c). EVI, SWC and rainfall were not significant in the model of dS/dt. The diurnal variation in dS/dt was greater at the mesic and xeric sites during the drought years 2010, 2011 and 2012, compared to the intermediate site, specifically during nighttime (SI Fig. S4). At the intermediate site dS/dt varied more during the years 2014 and 2016, as seen by greater entropy accumulation during nighttime hours and greater export during daytime hours for the year 2014.

## 4 Discussion

Here we describe differences in energy use efficiencies of sites with varying structural complexities (i.e., understory composition, basal area, DBH) using metrics of energy and entropy. Different from our expectations, environmental and structural effects on energy and entropy fluxes were not different with the exception of $NEE_e$ and $S_m$. These results suggest that differences in the thermodynamic environment among sites (i.e., air and surface temperatures) did not contribute to changes in entropy export in response to environmental variables. Metabolic entropy ($S_m$) decreased during the drought at all sites, but not significantly so (Fig. 7), whereas $NEE_e$ showed significant change at the mesic site. The different results were a function of SWC, which decreased during the summer of 2011, thus lowering the flux of $S_m$ (Fig. 7). Furthermore, greater

$R_{s,out}$ during the drought indicated lower available energy to drive photosynthetic processes. The decreases in $S_m$ and $NEE_e$ suggest that metabolic activity was affected by low rainfall, increasing VPD, and changes in temperature, demonstrating lower physiological activity of plant species during drought (Barron-Gafford et al., 2013). This decrease in metabolic efficiency supports a previous study at the mesic and xeric sites, which found lower electron transport and carboxylation capacity during drought (Wright et al., 2012).

Differences in the underlying reflective capacities at the sites significantly altered their entropy production and resulted in variation in entropy exchanges (Stoy et al., 2014). The more structurally complex mesic site had greater metabolic entropy production ($S_m$) compared with the intermediate site. Greater $S_m$ at the mesic site translates to greater energy accumulation, in addition to greater radiation entropy and export efficiencies ($eff_{rad}$, $eff_{flux}$) compared to the intermediate site, which had greater land use legacy and was structurally similar, but lower in plant functional diversity. Although the radiation entropy ratio ($eff_{rad}$) indicated that both the intermediate and xeric sites were equally energy efficient in terms of absorbing radiation, $eff_{flux}$ and $S_m$ showed prolonged recovery of energy efficiency from drought by one year at the intermediate site. Entropy change over time ($dS/dt$) did not significantly vary at the mesic site, but was more variable at the xeric and intermediate sites following the drought.

We hypothesized that the xeric site would have higher H and $J_H$, due to its open canopy and sandy soils and therefore lower volumetric heat capacity. In contrast to our first hypothesis, the mesic and intermediate sites and not the xeric site had a more pronounced increase in H and $J_H$ when EVI decreased during drought (Fig. 1). Lower H and $J_H$ at the xeric site was a consequence of greater energy partitioning into LE, enabled by greater transpiration rates of plant functional types present at the site (deciduous and evergreen oaks in the understory, mid- and overstory; Klein et al., 2013; Renninger et al., 2015; Stoy et al., 2006). This result was confirmed, as $J_H$ fluxes did not significantly change with an increase in EVI, whereas $J_{LE}$ increased, suggesting that evapotranspiration and the cooling of leaf and soil surfaces had greater influence on the partitioning of available energy. In contrast, $J_H$ increased more at the mesic and xeric sites with increasing VPD, suggesting that drier air increased the sensible heat flux from the surface to the atmosphere (Massmann et al., 2018). Similarly, as VPD increased so did σ at all sites. This response was also observed in Kuricheva et al. (2017), where drier summers resulted in greater entropy production, likely because an increase in VPD correlated with greater absorption of solar radiation and partitioning to H (Fig. 3a). Even though plant abundance was lower at the xeric site, its species composition was better adapted to drought conditions, which allowed for higher $J_{LE}$ compared to the other sites (Roman et al., 2015). Furthermore, an increase in EVI during summer months at the xeric site increased $J_{LE}$, demonstrating that greater leaf area enhanced ecosystem function (Peng et al., 2017; Zhu et al., 2016). Interestingly, $J_{LE}$ did not vary significantly by site with changes in VPD, which supported the findings of Whelan et al. (2013) that all sites had similar stomatal regulation to increases in VPD. Overall, the xeric site had higher $J_{LE}$ compared to the other sites for EVI < 0.5, even though the site basal area was almost half that of the mesic and intermediate sites (Table 1). An overstory composed of more oak species at the xeric site (~20 %) along with the $C_4$ understory resulted in higher transpiration during spring and summer, compared to stands containing just pine trees (Klein et al., 2013; Renninger et al., 2015; Stoy et al., 2006). Additionally, $C_4$ grasses and oak species at the xeric site were better adapted to drought (i.e., anisohydric response; Osborne and Sack, 2012; Roman et al., 2015), which may enable higher entropy production and lower variability in the

structural integrity (i.e., lower decreases in EVI; Fig. 1e). This suggests that the understory plays a crucial role in the structure and function of more open canopy ecosystems (Aoki, 2012; Lin, 2015), in addition to more productive overstory trees during summer. This led to similar entropy export efficiencies at all sites as evidenced by all sites having comparable dS/dt. Nevertheless, as σ increased with greater absorption of radiation due to an increase in EVI, $J_H$ decreased as a result of higher

SWC, resulting in temporary entropy accumulation at the xeric site during the end of 2012 (SI Fig. 4), which may have contributed to higher $T_{air}$ compared to the other sites (Fig. 2).

In contrast, the mesic site was affected by the interaction of biological and radiative forces, as $J_{LE}$ and $eff_{rad}$ decreased more severely with decreasing plant leaf area compared to the xeric site (lower EVI; Fig. 1e). As a consequence of lower LE and $J_{LE}$ during the drought, more energy was partitioned into H in 2011 (Fig. 6), as air, soil and surface temperatures increased due to

lower leaf area (Figs. 1 and 2), indicating a shift of ecosystem function (Ban-Weiss et al., 2011) towards lower quality energy degradation (Kuricheva et al., 2017). This initially depleted soil moisture storage at the mesic site (Fig. 1) and further decreased LE and $J_{LE}$ (Kim and Wang, 2012; Lauri et al., 2014). Nevertheless, the shift in energy partitioning at the mesic site allowed for the maintenance of dS/dt during drought, by export of entropy which had accumulated during nighttime hours (SI Fig. S4), demonstrating an adaptation of the site to changes in resource availability (Basu et al., 2016; Brodribb et al., 2014). In contrast,

the xeric and intermediate sites showed greater variability in annual dS/dt following the drought when rainfall returned to pre-drought levels and SWC increased (Fig. 10a). Nevertheless, the rapid increase in $J_{LE}$ in 2012 at the mesic and xeric sites indicated an increase in ecosystem function through greater evapotranspiration. This provides evidence of recovery following the drought, because $J_{LE}$ is of higher quality entropy dissipation (Kuricheva et al., 2017), coupling both mass and heat dynamics (Brunsell et al., 2011), whereas $J_H$ is a function of the thermal gradient (Kleidon, 2010; LeMone et al., 2007). In general, plant

species at the mesic site were better adapted to higher soil water conditions, as entropy and energy fluxes did not change as drastically with increasing SWC compared to the other sites.

This recovery of EVI following drought also allowed for greater $eff_{rad}$ at the sites. But $eff_{rad}$ was higher at the mesic site despite lower EVI compared to the intermediate site. This finding supports our second hypothesis, that sites with greater plant functional diversity maintain greater radiative entropy production. The mesic site efficiently used available energy from

incoming solar radiation (Fig. 2) through lower reflection of $R_s$ and by emitting less longwave radiation (Lin, 2015). $Eff_{rad}$ decreased during the initial drought year because all sites reflected more $R_s$, likely a consequence of a change in EVI, as well as leaf angle from a decrease in SWC and altered plant hydraulics. Higher $eff_{rad}$ and $eff_{flux}$ at the mesic site are consistent with enhanced function due to greater plant diversity in the understory (Fig. 4a). For example, wiregrass, a $C_4$ species, can maintain photosynthetic rates under high temperatures (Osborne and Sack, 2012; Ward et al., 1999), which allows for greater energy

storage during unfavorable environmental conditions (Brunsell et al., 2011). Despite higher wiregrass biomass in the understory, the xeric site was less efficient in using available radiation energy, indicated by high $R_{s,out}$ and $R_{l,out}$ (Brunsell et al., 2011). Structural limitations of the canopy (i.e., lower basal area), impeded the efficient absorption of available radiation, therefore lowering $eff_{rad}$ (Norris et al., 2011). Furthermore, larger proportions of deciduous oak trees at the xeric site (Table 1), which typically shed their leaves during the winter, lowered the capacity of the system to acquire radiation (Baldocchi et

al., 2004: Fig. 8b). Nevertheless, this inefficiency was not confirmed by model results for $S_m$, which, in contrast to $NEE_e$

revealed higher metabolic function at the xeric site relative to the mesic and intermediate sites, reflecting greater metabolic performance despite differences in basal area and site EVI. Overall our results demonstrate that the mesic site was better adapted to changes in resource availability by way of altering its reflective properties, where energy partitioning adjusted to maintain steady entropy exports relative to incoming entropy (Gunawardena et al., 2017; Otto et al., 2014; Taha et al., 1988).

Nevertheless, metabolic activity decreased during rainy periods ($S_m \sim 0$), demonstrating an inefficiency in maintaining optimal function when environmental pressure was imposed on the system. High metabolic function at the mesic site resulted in more rapid increases in the structural complexity as indicated by a decrease in $R_{s,out}$ following the drought when compared to the intermediate site (Brunsell et al., 2011; Holdaway et al., 2010). Metabolic activity (in energy terms) at the intermediate site was largely dependent on EVI (i.e., leaf area), demonstrating lower biological control of individual plant species (i.e., stomatal

control; Urban et al. 2016), but a strong influence of total leaf area on metabolic function and the export of entropy (Brunsell et al., 2011; Fig. 4 and 6). This was further illustrated at the intermediate site through less negative metabolic energy ($NEE_e$) when EVI was $\sim 0.25$ (Fig. 7c). Even though EVI in 2012 was greater at the intermediate site this did not correspond to higher $J_{LE}$ (Fig. 5a), which was also shown by a lack of significant change in entropy exports with changes in EVI ($eff_{flux}$, Fig. 9c). The result of lower metabolic function at the intermediate site is intriguing as the mesic and intermediate sites were structurally

similar, based on similar $B_A$, mean DBH and overstory tree composition (Table 1). The inefficiency appears to be a consequence of anthropogenic modification, which homogenized the ecosystem, leading to a decrease in understory plant functional types (Table. 1; Fig. 3), thereby reducing values of $eff_{rad}$, $eff_{flux}$ and $S_m$. This result provides evidence that the intermediate site was less efficient in absorbing energy and dissipating entropy compared to the mesic site, resulting in slower adaptation to drought. Similar results were shown in Lin et al. (2015), where disturbed sites had predominantly lower entropy

production rates, as well as in Lin et al. (2018) where greater surface temperature led to decreased $\sigma$, which we also observed at the intermediate site. Our third hypothesis was therefore supported, as the intermediate site had lower $eff_{flux}$ relative to the mesic and xeric sites. Lower plant functional diversity, specifically the lack of wiregrass, due to soil perturbations that took place prior to stand establishment (>95 years ago), likely lowered metabolic function, which in turn affected entropy exports at the intermediate site and its recovery from drought. For example, a negative $J_G$ at the intermediate site was observed with

increasing SWC suggesting poor soil water drainage, which is also likely a consequence of agricultural legacy (Kozlowski, 1999). A prolonged increase in $eff_{flux}$ compared to the other sites showed that the intermediate site did not adapt its entropy exports, in addition to greater reflection of $R_s$ during drought recovery. This result indicates that differences in soil conditions and lower plant functional diversity at the intermediate site reduced entropy exports compared to the other sites (Meysman and Bruers, 2010), such that plant functional types present at the site could not rescue the ecosystem's function during disturbance

(Elmqvist et al., 2003). Furthermore, while the intermediate site showed no change in dS/dt during the drought, following the drought the export of entropy significantly increased, resulting in more unstable conditions (Fig. 10a). The increase in entropy export corresponded to high annual rainfall and soil moisture conditions (Figs. 1 and S1), once more suggesting that soil characteristics were altered due to its agricultural legacy. The lower ability to adapt to changes in resource availability at the intermediate site could induce its degradation if environmental fluctuations, become more frequent and severe with climate

change (Mori, 2011; Siteur et al., 2016). This could further exacerbate instabilities for nearby sites, as changes in the reflective properties of degraded sites can alter microclimate and weather patterns across whole ecosystems (Norris et al., 2011).

We conclude that the analysis of entropy dynamics, in relation to structural and environmental variables gives valuable insights into the functional complexity of ecosystems and their ability to adapt to drought. A combination of entropy fluxes and entropy ratios revealed how differences in structural and/or functional characteristics affect energy efficiencies in longleaf pine ecosystems. Our results show that all sites demonstrated adaptive capacity to extreme drought, as indicated by a lack of significant change in dS/dt, except for greater variations at the xeric and intermediate sites following the drought. We show that overall low entropy exports at the site with greater land use legacy had the potential to decrease ecosystem function (Meysman and Bruers, 2010), especially during high rainfall events. Changes in climate and natural and human induced disturbances are becoming more frequent and severe (IPCC, 2014), demanding more predictive power about how changes in ecosystem structure and function will alter resilience to disturbances. Future policy, conservation or restoration applications depend on reliable measures such as the metrics presented here, to monitor ecosystem function following disturbances (Haddeland et al., 2014; Porter et al., 2012; Reinmann and Hutyra, 2016; Thom et al., 2017). This is especially critical for anthropogenically modified systems, as their land use history can affect changes in energy use efficiency and thus alter their ability to recover from disturbances (Bürgi et al., 2016; Foster et al., 2003). The application of entropy metrics could improve our understanding of the interaction of structure, function and legacy on energy use efficiency across a variety of global ecosystems.

**Author contribution**

G.S. and L.B. designed and acquired funding for the research. S.W. and C.S. analyzed the data. P.S. aided S.W. with the theories of entropy and energy density. All authors contributed to writing of the manuscript.

**Acknowledgement**

The authors thank the Forest Ecology laboratories personnel, with special thanks to Tanner Warren, Andres Baron-Lopez and Scott Taylor, for data collection and provision during the study at the Joseph W. Jones Ecological Research Center. CS and GS acknowledge support from the U.S. National Science Foundation (DEB EF-1241881). PS acknowledges support from the U.S. National Science Foundation (DEB 1552976, and 1702029) and the USDA National Institute of Food and Agriculture (Hatch project 228396).

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

**Table 1:** Stand characteristics at the mesic, intermediate and xeric sites at the Joseph W. Jones Ecological Research Center, Newton, GA, USA.

| Characteristic | Mesic | Intermediate | Xeric |
|---|---|---|---|
| Mean DBH (cm) | 25.9 | 42.5 | 22.5 |
| B$_A$ *P. palustris* (m$^2$ ha$^{-1}$) | 17.7 | 14.6 | 8.9 |
| B$_A$ all tree spp. (m$^2$ ha$^{-1}$) | 19.0 | 15.7 | 11.0 |
| Proportion of oak overstory trees (%) | 6.8 | 7.0 | 19.1 |
| LAI (m$^{-2}$ m$^{-2}$) | 1.0[a] | unknown | 0.69[a] |
| Wiregrass in the understory (%) | 28 | 5 | 24 |
| Woody species in the understory (%) | 12 | 15 | 10 |
| Prescribed fire | Early spring of 2009, 2011, 2013, 2015 | Early spring of 2009, 2011, 2013, 2015 | Early spring of 2009, 2011, 2013, 2015 |

a Wright et al. 2012

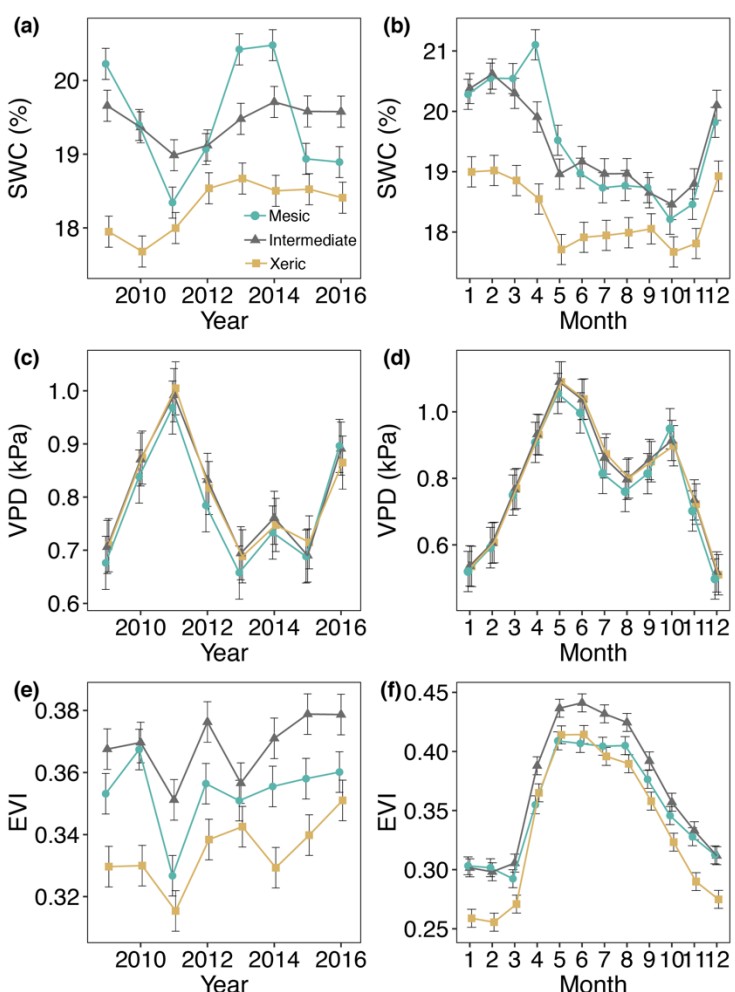

**Figure 1:** Least square mean predicted values from mixed models of environmental and structural variables for the years 2009-2016 at the mesic, intermediate and xeric sites, with average annual (a, c, and e) and monthly (b, d, and f) means of (a and b) soil water content (SWC), (c and d) vapor pressure deficit (VPD), and (e and f) enhanced vegetation index (EVI). Error bars represent standard errors (SE).

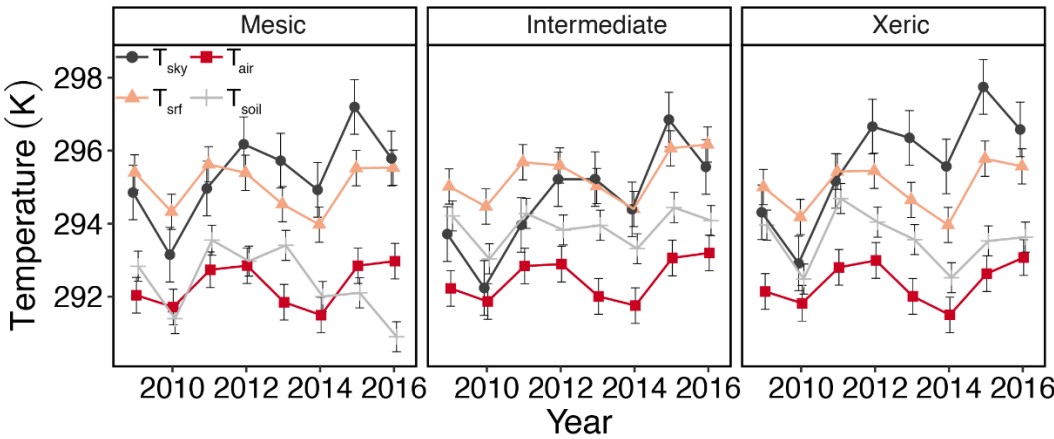

**Figure 2:** Least square mean predicted values from mixed models of annual sky temperature ($T_{sky}$), air temperature ($T_{air}$), surface temperature ($T_{srf}$), and soil temperature ($T_{soil}$) at the mesic, intermediate and xeric sites. Error bars represent SE.

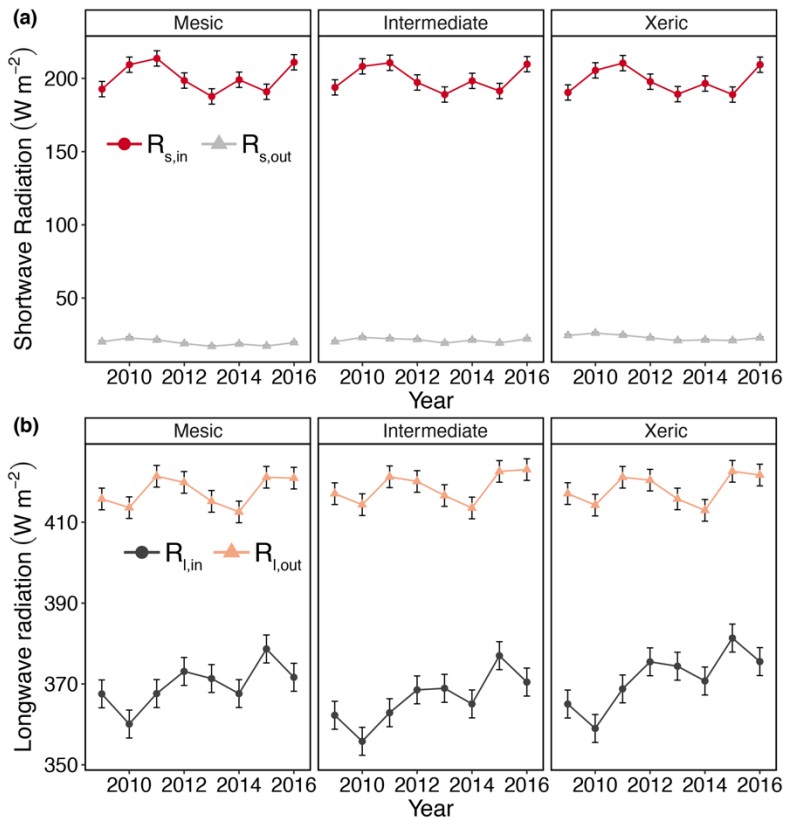

**Figure 3:** Least square mean predicted values from mixed models of annual average radiation at the mesic, intermediate and xeric sites for the years 2009-2016: (a) annual incoming and outgoing shortwave radiation ($R_{s,in}$ and $R_{s,out}$), and (b) annual incoming and outgoing longwave radiation ($R_{l,in}$ and $R_{l,out}$). Error bars represent SE.

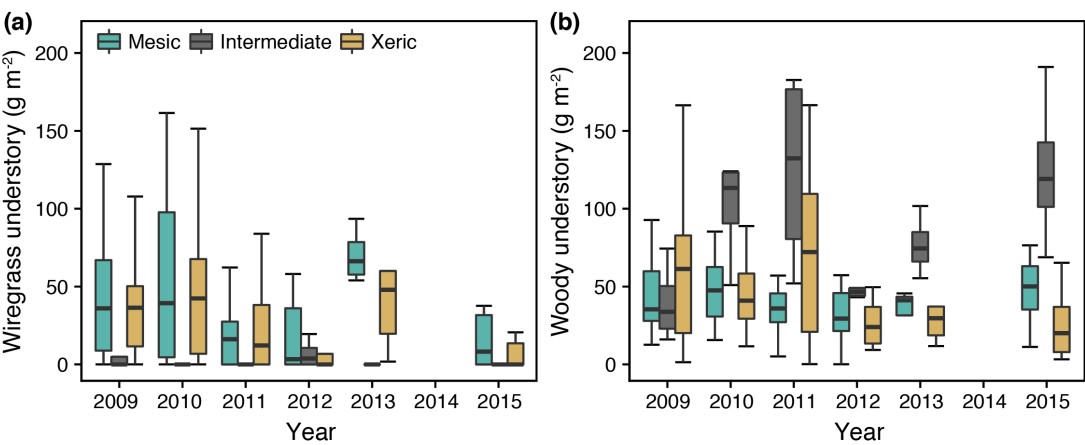

**Figure 4:** (a) Wiregrass and (b) woody understory biomass from 2009 through 2015 at the mesic, intermediate and xeric sites. Note that the sampling protocol changed to a 2-year measurements cycle in 2013, such that measurements were not made in 2014 and 2016.

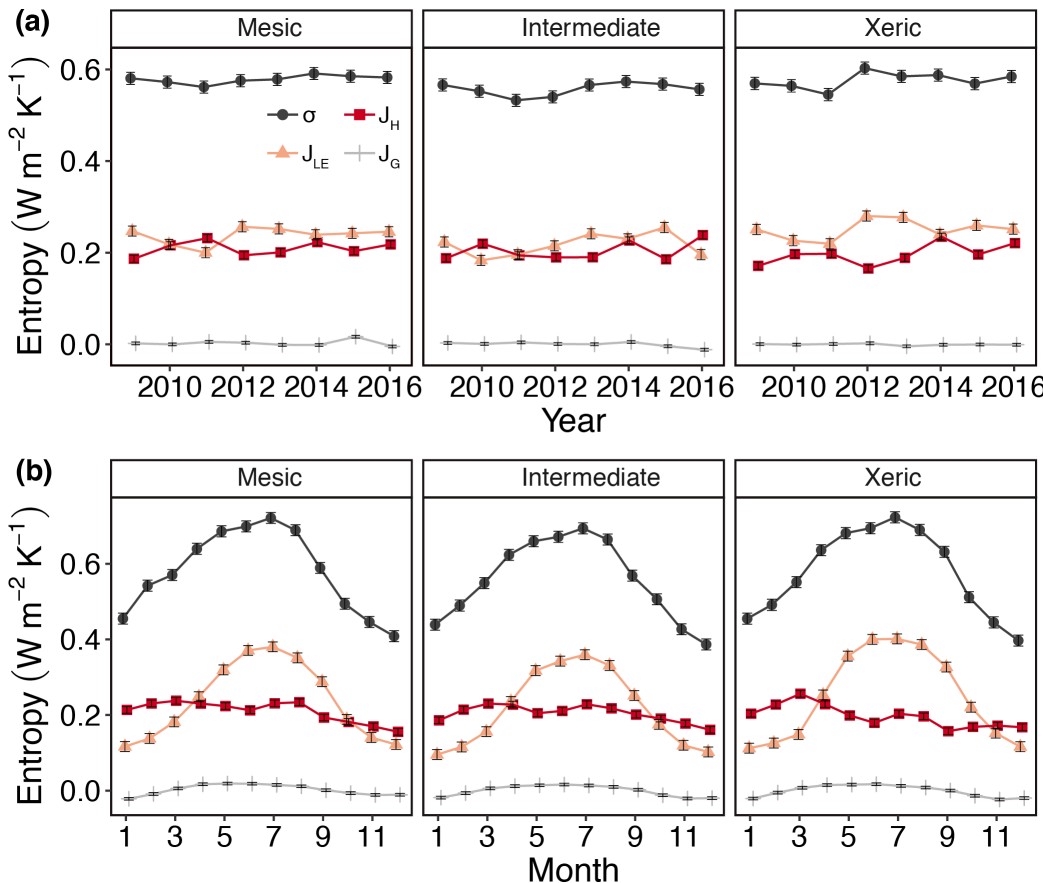

**Figure 5:** Least square mean predicted values from mixed models of annual (a) and monthly (b) average entropy production ($\sigma$) and entropy fluxes of latent energy ($J_{LE}$), sensible heat ($J_H$), and ground heat ($J_G$) at the mesic, intermediate and xeric sites. Error bars represent SE.

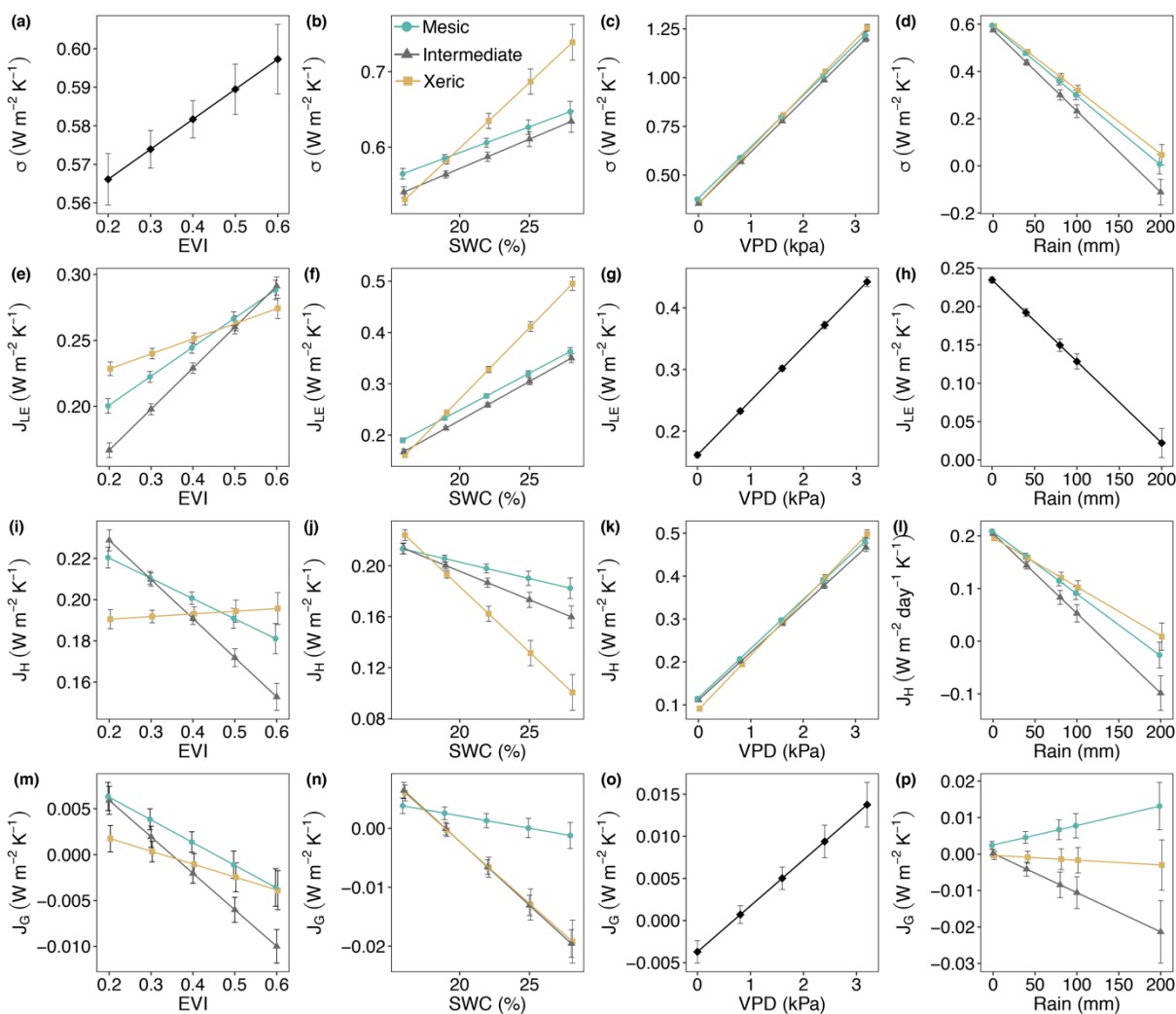

**Figure 6:** Least square mean predicted values from mixed models of (a-d) entropy production (σ) and entropy fluxes of (e-h) latent energy (J$_{LE}$), (i-l) sensible heat (J$_H$), and (m-p) ground heat (J$_G$) by site and (a, e, i, m) enhanced vegetation index (EVI), (b, f, j, n) soil water content (SWC), (c, g, k, o) vapor pressure deficit (VPD), and (d, h, l, p) rain. For (g), (h) and (o) the interaction with site was not significant, as signified by a single black line. Error bars represent SE.

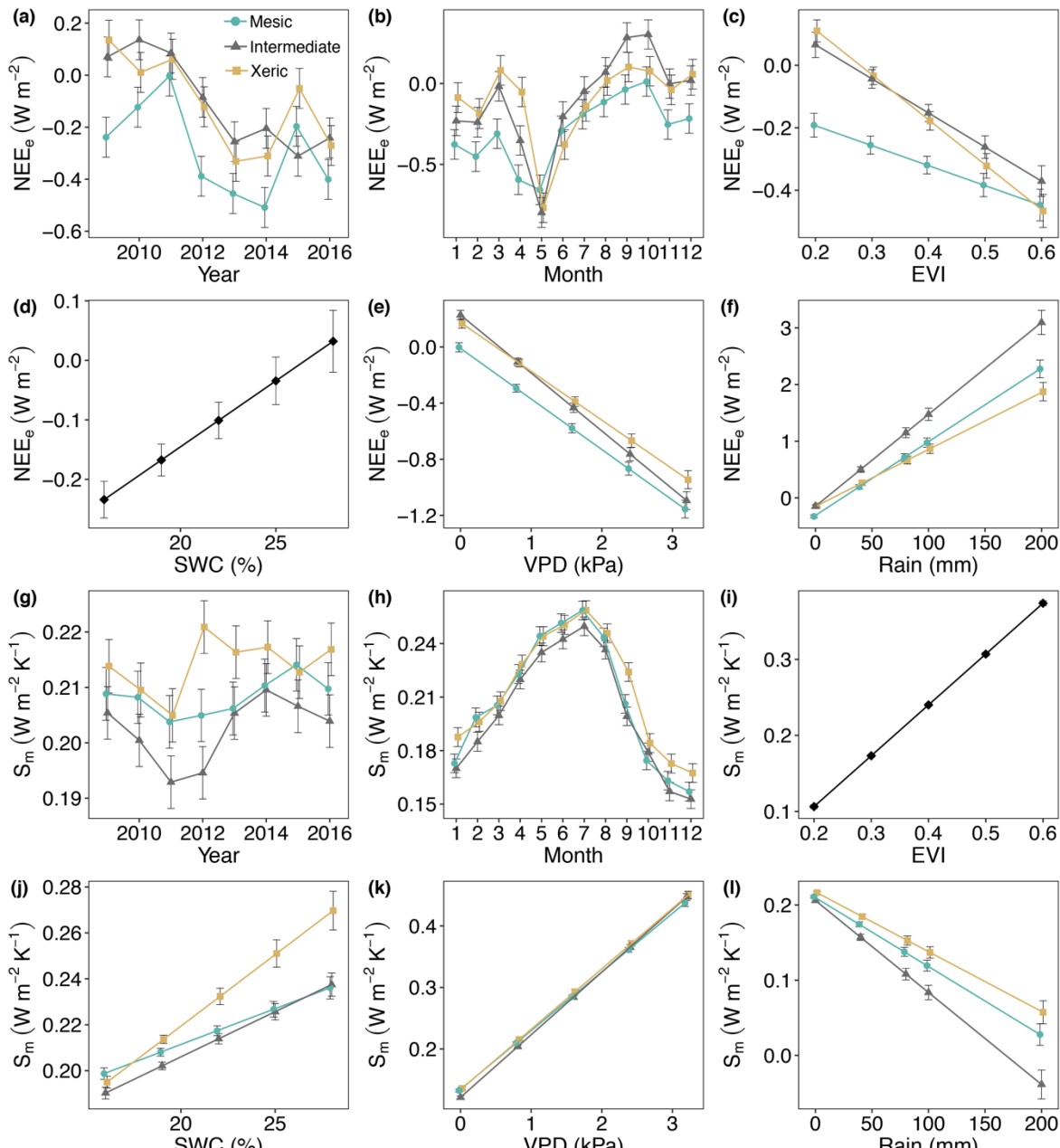

**Figure 7:** Least square mean predictive values from mixed model of (a-f) the metabolic energy flux (NEE_e) and (g and l) metabolic entropy fluxes of (S_m) by site and (a and g) year, (b and h) month, (c and i) enhanced vegetation index (EVI), (d and j) soil water content (SWC), (e and k) vapor pressure deficit (VPD), and (f and l) rain. For (d) and (i) the interaction with site was not significant, as indicated by a single solid black line. Error bars represent SE.

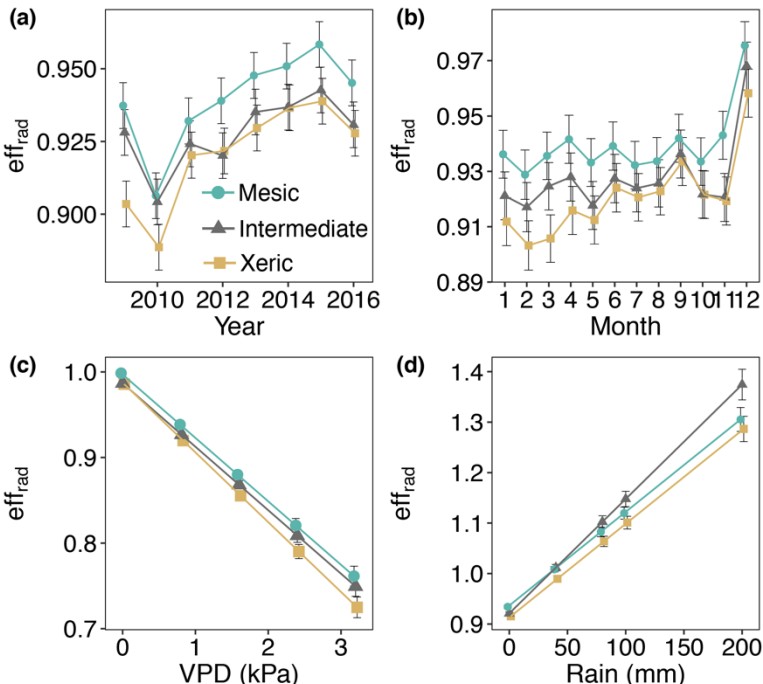

**Figure 8:** Least square mean predicted values from mixed models of average daily half-hourly radiative entropy efficiencies (eff$_{rad}$) at the mesic, intermediate and xeric sites by (a) year, (b) month, (c) vapor pressure deficit (VPD), and (d) rain. Soil water content and the enhanced vegetation index were not significant in the model. Error bars represent SE.

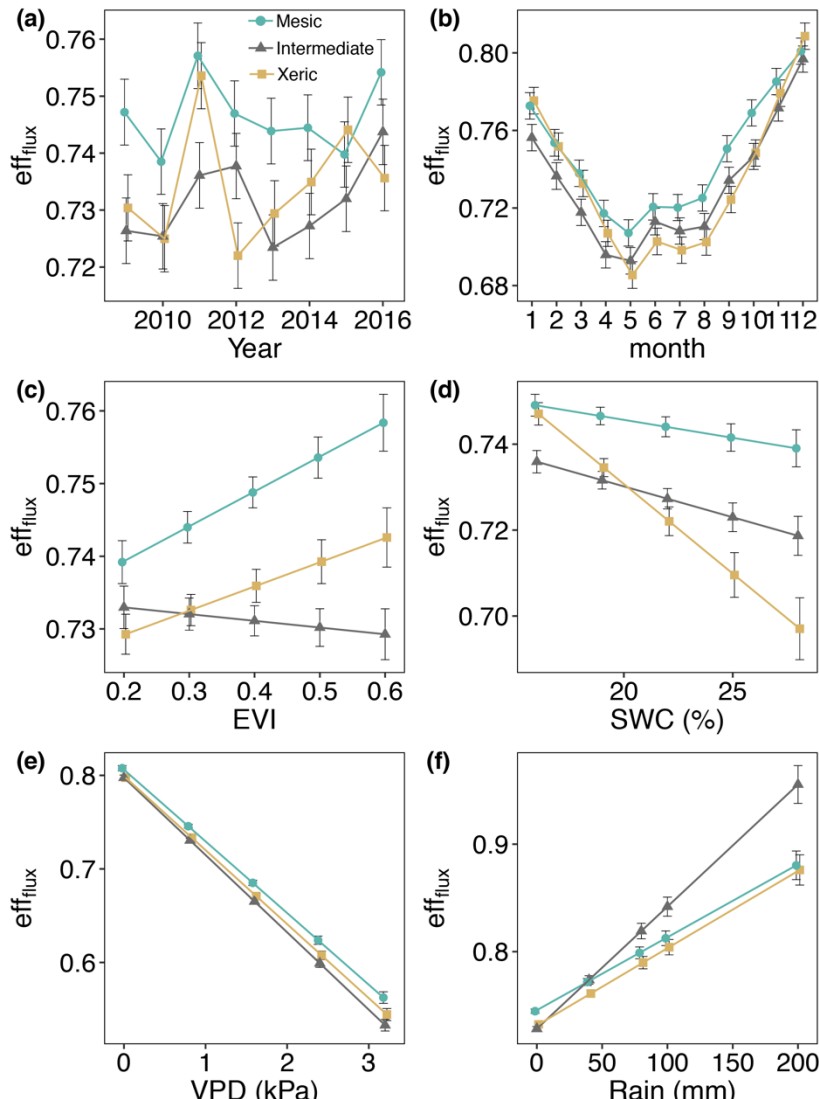

**Figure 9:** Least square mean predicted values from mixed models of average daily half-hourly flux entropy efficiencies (eff$_{flux}$) at the mesic, intermediate and xeric sites by (a) year, (b) month, (c) enhanced vegetation index (EVI), (d) soil water content (SWC), (e) vapor pressure deficit (VPD, and (f) rain. Error bars represent SE.

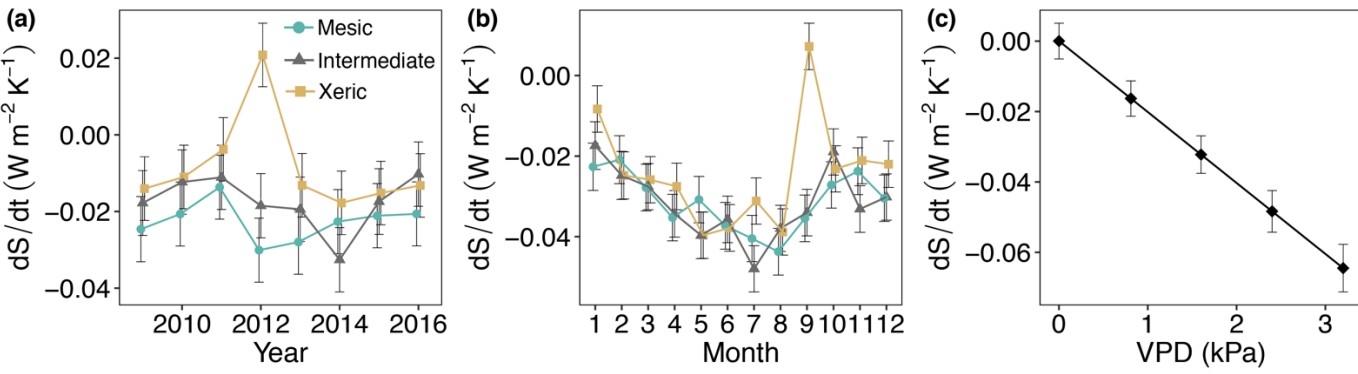

**Figure 10:** Least square mean predicted values from mixed models of average daily entropy at the mesic, intermediate and xeric sites by (a) year (a) month, (c) vapor pressure deficit (VPD). Soil water content and rain, as well as the interactions with site were not significant in the model. Error bars represent SE.

"Quantifying energy use efficiency via entropy production: A case study from longleaf pine ecosystems"

**Table S1:** Type 3 tests of fixed effects for model of rain.

| Effect | Sum Sq. | Df | F value | Pr(>F) |
|--------|---------|-----|---------|--------|
| Site | 30607 | 2 | 4.2059 | 0.0159 |
| Year | 95938 | 7 | 3.7667 | < 0.001 |

**Table S2:** Type 3 tests of fixed effects for the models of environmental variables and radiation.

| Model | Effect | Chisq | Df | p-value |
|---|---|---|---|---|
| SWC | Site | 6561.692 | 2 | < 0.001 |
| | Year | 23.764 | 7 | 0.0013 |
| | Month | 94.089 | 11 | < 0.001 |
| | Site:Year | 2629.617 | 14 | < 0.001 |
| | Site:Month | 1398.986 | 22 | < 0.001 |
| VPD | Site | 245.268 | 2 | < 0.001 |
| | Year | 33.981 | 7 | < 0.001 |
| | Month | 100.044 | 11 | < 0.001 |
| | Site:Year | 214.101 | 14 | < 0.001 |
| | Site:Month | 232.327 | 22 | < 0.001 |
| EVI | Site | 2510.727 | 2 | < 0.001 |
| | Year | 15.868 | 7 | 0.0264 |
| | Month | 597.701 | 11 | < 0.001 |
| | Site:Year | 294.805 | 14 | < 0.001 |
| | Site:Month | 791.727 | 22 | < 0.001 |
| $T_{sky}$ | Site | 2202.369 | 2 | < 0.001 |
| | Year | 23.089 | 7 | 0.0017 |
| | Month | 912.141 | 11 | < 0.001 |
| | Site:Year | 440.318 | 14 | < 0.001 |
| | Site:Month | 63.082 | 22 | < 0.001 |
| $T_{srf}$ | Site | 438.625 | 2 | < 0.001 |
| | Year | 12.844 | 7 | 0.076 |
| | Month | 1423.846 | 11 | < 0.001 |
| | Site:Year | 435.639 | 14 | < 0.001 |
| | Site:Month | 778.064 | 22 | < 0.001 |
| $T_{air}$ | Site | 1419.775 | 2 | < 0.001 |
| | Year | 9.954 | 7 | 0.1912 |
| | Month | 1231.11 | 11 | < 0.001 |
| | Site:Year | 1311.82 | 14 | < 0.001 |
| | Site:Month | 336.866 | 22 | < 0.001 |
| $T_{soil}$ | Site | 5110.24 | 2 | < 0.001 |
| | Year | 16.817 | 7 | 0.0186 |
| | Month | 1901.818 | 11 | < 0.001 |
| | Site:Year | 1922.717 | 14 | < 0.001 |
| | Site:Month | 5270.008 | 22 | < 0.001 |
| $R_{s,in}$ | Site | 0.9664 | 2 | 0.6168 |
| | Year | 16.3199 | 7 | 0.0224 |
| | Month | 763.0665 | 11 | < 0.001 |
| | Site:Year | 121.9389 | 14 | < 0.001 |
| | Site:Month | 170.75 | 22 | < 0.001 |
| $R_{s,out}$ | Site | 4161.151 | 2 | < 0.001 |
| | Year | 48.782 | 7 | < 0.001 |
| | Month | 682.874 | 11 | < 0.001 |
| | Site:Year | 816.733 | 14 | < 0.001 |
| | Site:Month | 1780.397 | 22 | < 0.001 |
| $R_{l,in}$ | Site | 2479.339 | 2 | < 0.001 |
| | Year | 22.578 | 7 | 0.0020 |
| | Month | 1005.462 | 11 | < 0.001 |
| | Site:Year | 482.99 | 14 | < 0.001 |
| | Site:Month | 72.965 | 22 | < 0.001 |
| $R_{l,out}$ | Site | 226.43 | 2 | < 0.001 |
| | Year | 13.07 | 7 | 0.0704 |
| | Month | 1433.87 | 11 | < 0.001 |
| | Site:Year | 137.39 | 14 | < 0.001 |
| | Site:Month | 980.18 | 22 | < 0.001 |

**Table S3:** Type 3 tests of fixed effects for the models of energy.

| Model | Effect | Chisq. | Df | Pr(>Chisq) |
|-------|--------|--------|----|-----------|
| **$R_n$** | Year | 20.6658 | 7 | 0.0042975 |
| | Month | 1927.222 | 11 | < 0.001 |
| | SWC | 58.6889 | 1 | < 0.001 |
| | Site | 650.5143 | 2 | < 0.001 |
| | EVI | 12.2151 | 1 | 0.0005 |
| | Rain | 140.9816 | 1 | < 0.001 |
| | VPD | 1756.8922 | 1 | < 0.001 |
| | Month:Site | 120.9114 | 22 | < 0.001 |
| | SWC:Site | 24.2945 | 2 | < 0.001 |
| | Site:EVI | 7.3321 | 2 | 0.0256 |
| | Site:VPD | 16.6743 | 2 | 0.0002 |
| | Year:Site | 263.8642 | 14 | < 0.001 |
| **LE** | Year | 20.7768 | 7 | 0.0041 |
| | Month | 754.2793 | 11 | < 0.001 |
| | SWC | 455.4372 | 1 | < 0.001 |
| | Site | 476.4295 | 2 | < 0.001 |
| | EVI | 149.9341 | 1 | < 0.001 |
| | Rain | 116.5615 | 1 | < 0.001 |
| | VPD | 1043.0314 | 1 | < 0.001 |
| | Month:Site | 369.8495 | 22 | < 0.001 |
| | SWC:Site | 130.9093 | 2 | < 0.001 |
| | Site:EVI | 43.0759 | 2 | < 0.001 |
| | Site:VPD | 5.3897 | 2 | 0.0676 |
| | Year:Site | 564.6937 | 14 | < 0.001 |
| **H** | Year | 39.525 | 7 | < 0.001 |
| | Month | 108.742 | 11 | < 0.001 |
| | SWC | 29.086 | 1 | < 0.001 |
| | Site | 90.131 | 2 | < 0.001 |
| | EVI | 25.974 | 1 | < 0.001 |
| | Rain | 95.918 | 1 | < 0.001 |
| | VPD | 1320.893 | 1 | < 0.001 |
| | Month:Site | 301.757 | 22 | < 0.001 |
| | SWC:Site | 35.234 | 2 | < 0.001 |
| | Site:EVI | 41.862 | 2 | < 0.001 |
| | Site:VPD | 29.24 | 2 | < 0.001 |
| | Site:Rain | 16.416 | 2 | 0.0003 |
| | Year:Site | 351.685 | 14 | < 0.001 |
| **G** | Year | 9.1742 | 7 | 0.2404 |
| | Month | 180.4785 | 11 | < 0.001 |
| | SWC | 37.8658 | 1 | < 0.001 |
| | Site | 200.7208 | 2 | < 0.001 |
| | EVI | 33.4003 | 1 | < 0.001 |
| | Rain | 0.1512 | 1 | 0.6974 |
| | VPD | 36.7781 | 1 | < 0.001 |
| | Month:Site | 375.8069 | 22 | < 0.001 |
| | SWC:Site | 38.7949 | 2 | < 0.001 |
| | Site:EVI | 8.2576 | 2 | 0.0161 |
| | Site:Rain | 14.6424 | 2 | 0.0007 |
| | Site:VPD | 6.4624 | 2 | 0.0395 |
| | Year:Site | 990.9702 | 14 | < 0.001 |

**Table S4:** Type 3 tests of fixed effects for models of entropy.

| Model | Effect | Chisq | Df | Pr(>Chisq) |
|---|---|---|---|---|
| $\sigma$ | Year | 4.8186 | 7 | 0.6821 |
| | Month | 615.1107 | 11 | < 0.001 |
| | SWC | 37.0574 | 1 | < 0.001 |
| | Site | 146.6553 | 2 | < 0.001 |
| | EVI | 6.1264 | 1 | 0.0133 |
| | Rain | 247.8162 | 1 | < 0.001 |
| | VPD | 2170.749 | 1 | < 0.001 |
| | Month:Site | 156.0447 | 22 | < 0.001 |
| | SWC:Site | 22.0878 | 2 | < 0.001 |
| | Site:VPD | 10.2957 | 2 | 0.0058 |
| | Site:Rain | 9.0465 | 2 | 0.0109 |
| | Year:Site | 117.8733 | 14 | < 0.001 |
| $J_{LE}$ | Year | 21.216 | 7 | 0.0035 |
| | Month | 726.81 | 11 | < 0.001 |
| | SWC | 456.76 | 1 | < 0.001 |
| | Site | 493.661 | 2 | < 0.001 |
| | EVI | 148.839 | 1 | < 0.001 |
| | Rain | 127.775 | 1 | < 0.001 |
| | VPD | 1011.278 | 1 | < 0.001 |
| | Month:Site | 367.42 | 22 | < 0.001 |
| | SWC:Site | 162.581 | 2 | < 0.001 |
| | Site:EVI | 42.076 | 2 | < 0.001 |
| | Year:Site | 560.321 | 14 | < 0.001 |
| $J_H$ | Year | 38.625 | 7 | < 0.001 |
| | Month | 101.071 | 11 | < 0.001 |
| | SWC | 25.483 | 1 | < 0.001 |
| | Site | 93.504 | 2 | < 0.001 |
| | EVI | 25.804 | 1 | < 0.001 |
| | Rain | 94.524 | 1 | < 0.001 |
| | VPD | 1208.397 | 1 | < 0.001 |
| | Month:Site | 315.446 | 22 | < 0.001 |
| | SWC:Site | 39.127 | 2 | < 0.001 |
| | Site:EVI | 44.953 | 2 | < 0.001 |
| | Site:VPD | 30.372 | 2 | < 0.001 |
| | Site:rRain | 14.251 | 2 | 0.0008 |
| | Year:Site | 370.91 | 14 | < 0.001 |
| $J_G$ | Year | 7.6197 | 7 | 0.3673 |
| | Month | 180.1628 | 11 | < 0.001 |
| | SWC | 35.1066 | 1 | < 0.001 |
| | Site | 234.691 | 2 | < 0.001 |
| | EVI | 31.1994 | 1 | < 0.001 |
| | Rain | 0.8563 | 1 | 0.3548 |
| | VPD | 29.1953 | 1 | < 0.001 |
| | Month:Site | 299.2461 | 22 | < 0.001 |
| | SWC:Site | 56.2234 | 2 | < 0.001 |
| | Site:EVI | 11.0306 | 2 | 0.004 |
| | Site:Rain | 22.1752 | 2 | < 0.001 |
| | Year:Site | 1082.405 | 14 | < 0.001 |

**Table S5:** Type 3 tests of fixed effects for models of metabolic energy ($NEE_e$) and entropy ($S_m$).

| Effect | Chisq | Df | Pr(>Chisq) | Effect |
|--------|-------|-----|------------|--------|
| **NEE$_e$** | Year | 29.646 | 7 | 0.0001102 |
| | Month | 74.127 | 11 | < 0.001 |
| | SWC | 19.826 | 1 | < 0.001 |
| | Site | 779.838 | 2 | < 0.001 |
| | EVI | 75.114 | 1 | < 0.001 |
| | Rain | 300.884 | 1 | < 0.001 |
| | VPD | 327.07 | 1 | < 0.001 |
| | Month:Site | 742.229 | 22 | < 0.001 |
| | Site:EVI | 14.519 | 2 | 0.0007 |
| | Site:VPD | 11.067 | 2 | 0.0034 |
| | Site:Rain | 42.48 | 2 | < 0.001 |
| | Year:Site | 520.107 | 14 | < 0.001 |
| **S$_m$** | Year | 4.1562 | 7 | 0.7616 |
| | Month | 475.1864 | 11 | < 0.001 |
| | SWC | 63.6594 | 1 | < 0.001 |
| | Site | 138.5192 | 2 | < 0.001 |
| | EVI | 3580.1388 | 1 | < 0.001 |
| | Rain | 189.1466 | 1 | < 0.001 |
| | VPD | 2433.3156 | 1 | < 0.001 |
| | Month:Site | 129.5421 | 22 | < 0.001 |
| | SWC:Site | 13.7285 | 2 | 0.001 |
| | Site:VPD | 12.6944 | 2 | 0.0002 |
| | Site:Rain | 26.3357 | 2 | < 0.001 |
| | Year:Site | 118.2567 | 14 | < 0.001 |

**Table S6:** Type 3 tests of fixed effects for models of entropy efficiency.

| Model | Effect | Chisq | Df | Pr(>Chisq) |
|---|---|---|---|---|
| **eff$_{rad}$** | Site | 351.3632 | 2 | < 0.001 |
| | Year | 26.8439 | 7 | 0.0004 |
| | Month | 26.4847 | 11 | 0.0055 |
| | VPD | 282.0899 | 1 | < 0.001 |
| | Rain | 282.8825 | 1 | < 0.001 |
| | Site:Month | 74.9708 | 22 | < 0.001 |
| | Site:VPD | 9.8219 | 2 | 0.0074 |
| | Site:Rain | 17.4162 | 2 | 0.0001 |
| | Site:Year | 121.4967 | 14 | < 0.001 |
| **eff$_{flux}$** | Mite | 938.8639 | 2 | < 0.001 |
| | Year | 9.2791 | 7 | 0.2332 |
| | Month | 251.1215 | 11 | < 0.001 |
| | VPD | 1204.1726 | 1 | < 0.001 |
| | EVI | 5.4535 | 1 | 0.0195 |
| | Rain | 122.5276 | 1 | < 0.001 |
| | SWC | 8.9111 | 1 | 0.0028 |
| | Site:Month | 307.582 | 22 | < 0.001 |
| | Site:SWC | 25.8864 | 2 | < 0.001 |
| | Site:VPD | 17.4305 | 2 | 0.0002 |
| | Site:Rain | 51.4031 | 2 | < 0.001 |
| | Site:EVI | 15.1919 | 2 | 0.0005 |
| | Site:Year | 517.3889 | 14 | < 0.001 |
| **dS/dt** | Site | 31.8302 | 2 | < 0.001 |
| | Year | 5.4413 | 7 | 0.6063 |
| | Month | 30.5512 | 11 | 0.0013 |
| | VPD | 139.6233 | 1 | < 0.001 |
| | Site:Month | 100.2884 | 22 | < 0.001 |
| | Site:Year | 124.8501 | 14 | < 0.001 |

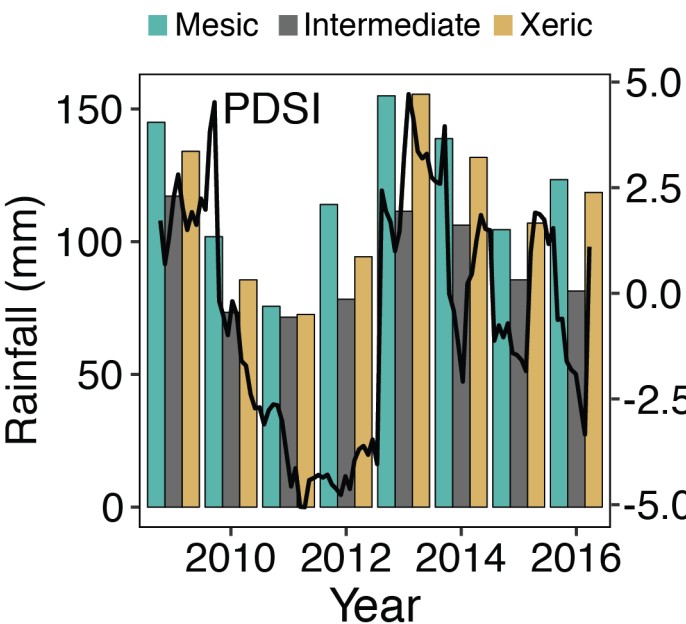

**Figure S1:** Monthly rainfall sums and Palmer Drought Severity Index (PDSI) for the mesic, intermediate and xeric sites from 2009 through 2016.

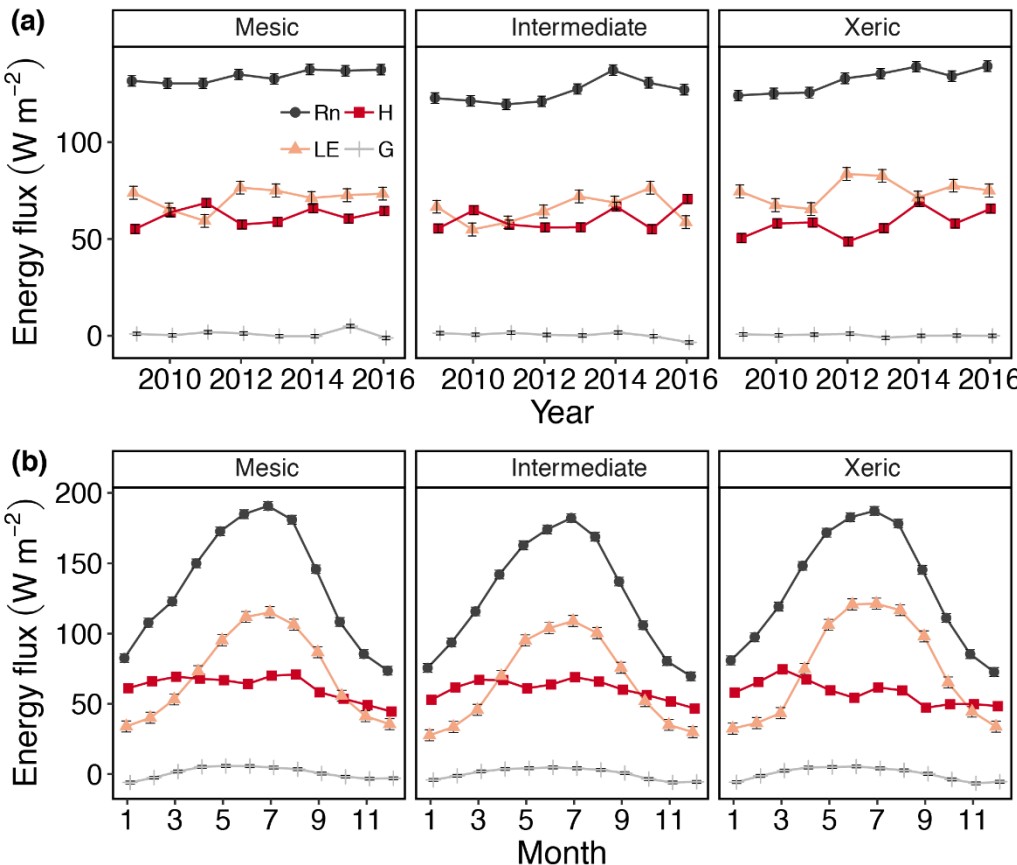

**Figure S2:** Annual (a) and monthly (b) changes of the energy fluxes of net radiation ($R_n$), latent energy (LE), sensible heat (H), and ground heat (G) at the mesic, intermediate and xeric sites.

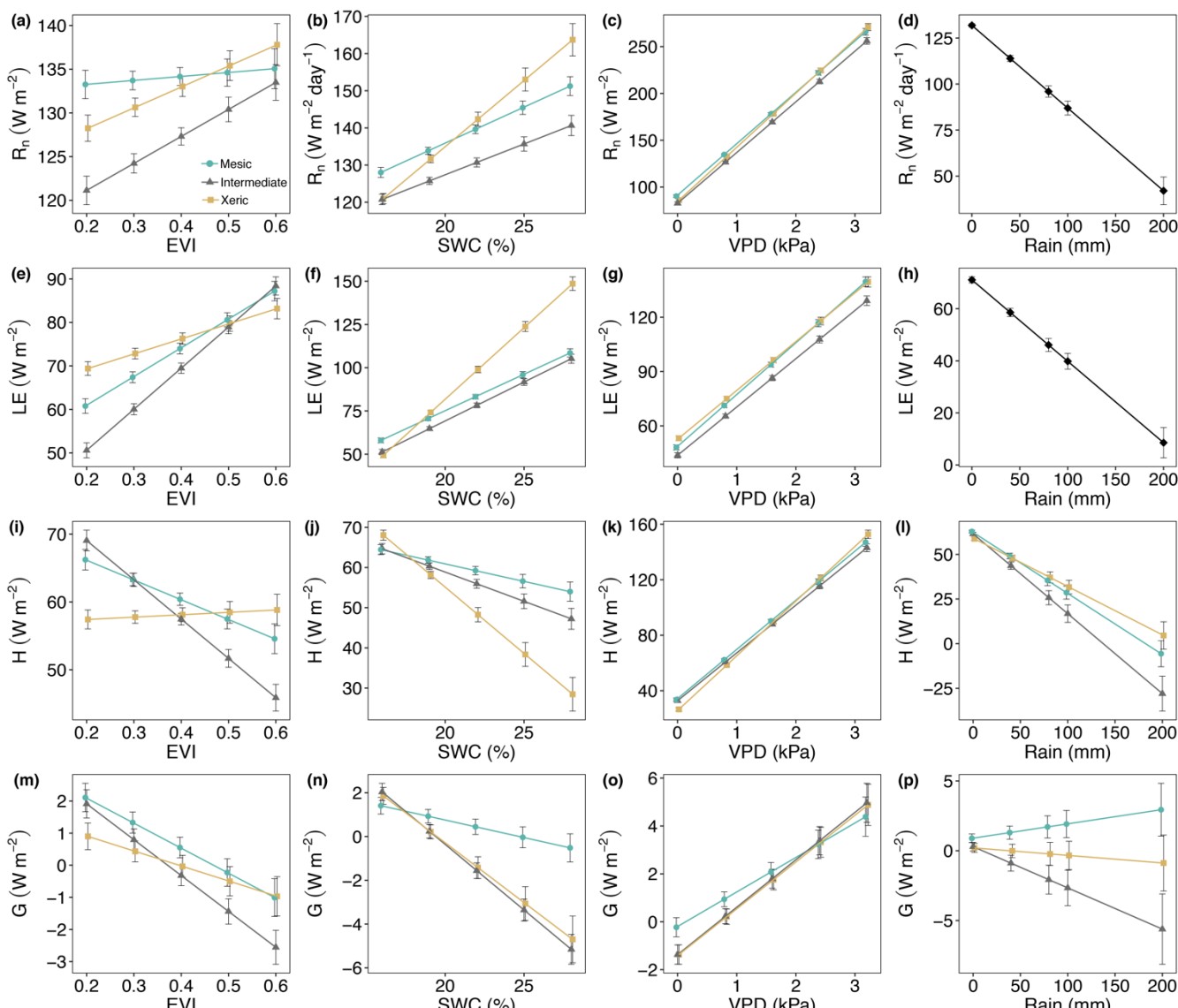

**Figure S3:** Least square mean predicted values from mixed models of energy fluxes of (a-d) net radiation ($R_n$), (e-h) latent energy (LE), (i-l) sensible heat (H), and (m-p) ground heat (G) by site and (a, e, i, m) enhanced vegetation index (EVI), (b, f, j, n) soil water content (SWC), (c, g, k, o) vapor pressure deficit (VPD) and (d, h, l, p) rain. For (d) and (h) the interaction with site was not significant, as indicated by a single solid black line.

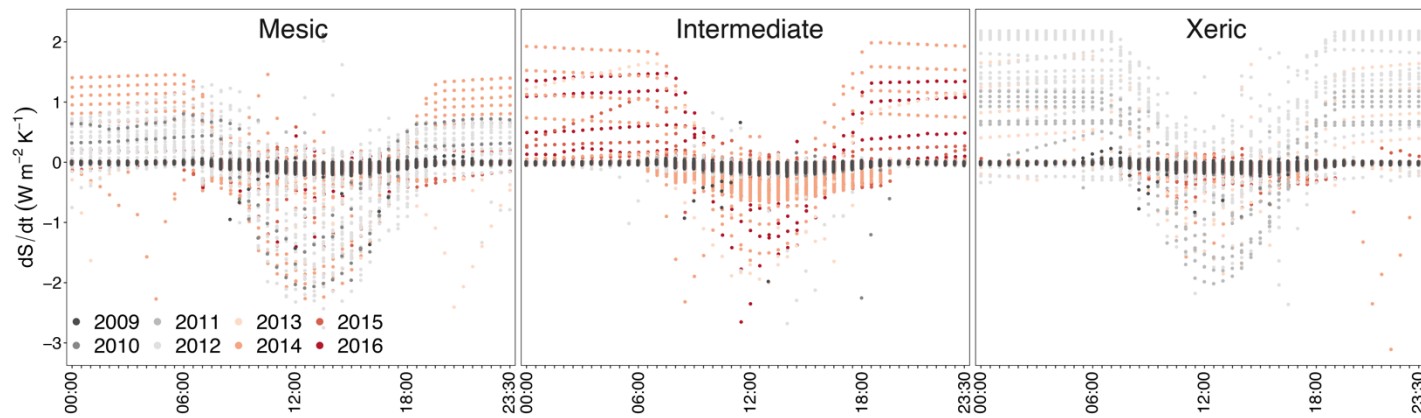

**Figure S4:** Diurnal changes in entropy (dS/dt) at the mesic, intermediate and xeric sites for the years 2011 through 2016.