# Peer review of "Quantifying energy use efficiency via entropy production: A case study from longleaf pine ecosystems"

_Biogeosciences, 2018_

## Referee Comment (RC1) · A. Kleidon (Referee) · 15 Sep 2018

Summary:

This manuscript describes a study in which entropy production is used to assess the resilience of ecosystems. To do so, this study diagnoses entropy fluxes and entropy production and describes these for three observational sites. In principle, I think that the approach is certainly unusual and novel. Yet, they way that entropy fluxes and production are diagnosed shows some major deficiencies which will certainly impact the results their interpretation. Also, I am not at all convinced that there is extra insights gained by looking at entropy production rather than a conventional analysis of

the energy, water, and carbon balances. Hence, I cannot recommend publication in this form.

Major comments:

1. Diagnosis of entropy fluxes and entropy production

I think that there are some major flaws in the methodology of how entropy fluxes and entropy production are being diagnosed. I understand that the authors use formulations from previously published papers, yet, as I will describe, I think that these are incorrect.

First, the entropy balance is used in Eq. 9, stating that the "overall change in entropy production (S) over time (t) in kJ m-2 K-1 of the ecosystem [is estimated] by adding entropy flux and entropy production". This is incorrect. What Eq. 9 formulates is the entropy balance. It balances the change in entropy on the left hand side of the equation (dS/dt) with the sum of all entropy exchange fluxes (J) and all entropy production terms ($\sigma$). This balance is typically assumed to be zero in a steady state, i.e., dS/dt = 0, which then allows one to diagnose entropy production from the difference in entropy exchange fluxes. This is in fact what the authors do to diagnose entropy production in Eqs. 3.6 and 3.7 to diagnose entropy production by absorption of radiation. Yet, the authors later use dS/dt in Eq. 4.8 to derive an efficiency. This efficiency should be zero, otherwise they did not do the balancing correctly. So there is a major inconsistency in the methodology that needs to be resolved.

Second, entropy production by absorption of longwave radiation is estimated using net longwave radiation at the surface (Eq. 3.7). What is the justification for using net longwave radiation, rather than gross fluxes? After all, the downwelling longwave radiation of the surface adds an entropy flux of Rldown/Tsky, while the emission of radiation from the surface exports entropy at the rate of Rlup/Tsrf. Using the difference of these two fluxes (assuming that dS/dt=0) yields an entropy production of $\sigma$ = Rlup/Tsrf - Rldown/Tsky, which is not the same as (Rlup - Rldown) * (1/Tsrf - 1/Tsky). The authors should correct this, or explain why their expression is justified.

The same reasoning applies to the application of net ecosystem exchange, where I think that also gross fluxes should be used, not net fluxes.

2. Additional insights gained from entropy fluxes and entropy production

The authors link their entropy-based analysis to rather general concepts such as resilience and energy use efficiency. Yet, I do not see the additional insights gained by using entropy production, rather than an analysis based on the entropy, water, and carbon balance. Why does the entropy-based analysis provide more or novel insights that cannot be obtained by just an interpretation based on fluxes? The authors do not really answer this question within the manuscript and do not use the results to show this, as they only focus on an entropy-based analysis.

In terms of interpreting the observations, I think that there is a critical step missing that relates the observed differences to an interpretation of processes, and this cannot be gained by just looking at entropy. For instance, temperature changes result from changes in the energy balance, as temperature is a measure for heat content. Yet, the energy balance is not even shown or discussed. Likewise, to understand changes in evaporation, I would expect a water balance being discussed. Instead, this study directly diagnoses entropy fluxes and thereby skips this process-based level of interpretation. It does not show and interpret the fluxes of the energy, water, and carbon balances separately, and does not demonstrate that something else can be learned by looking at entropy.

By lumping all aspects of the land surface into entropy production, I think that this neglects those aspects that are relevant for ecosystems from those that are irrelevant. The relevant flux for ecosystems is primarily the uptake of carbon, as this provides the chemical energy for terrestrial ecosystems. Plants live from the energy they fix during carbon assimilation, and, quite frankly, care little about the entropy production of other processes.

For this manuscript to provide more solid insights, I think it needs a more processbased interpretation using the available data, it needs to be more specific regarding those terms that are really relevant to ecosystems, and it needs to at least discuss why there is more to be gained by looking at entropy-based diagnostics.

Minor comments:

Abstract: "Our study provides foundational evidence of how MEP can be used to determine resiliency across ecosystems globally" - I am not at all convinced and doubt this conclusion. The authors provide no discussion why a diagnosis based on entropy fluxes yields more or better insights than the diagnosis of energy, water, and carbon balances. I see this as a critical missing bit in this manuscript.

Introduction, page 2, line 16: MEP is referred to as a principle in the text. At best, it is a "proposed" principle, or better hypothesis, as it is not generally being accepted.

page 2, line 24: How can agricultural systems exceed MEP if MEP already describes the maximum? This does not make sense. What I can imagine is that agricultural systems maintain a different state because of nutrient inputs, but then, the boundary conditions are changed because there are additional exchange fluxes across the system boundary. Also, why would this excessive entropy production be unsustainable? As long as the nutrient input can be maintained, I see no reason why it should be unsustainable.

page 3, line 9: What are entropy efficiency ratios? In thermodynamics, efficiency is used to describe the conversion efficiency of one form of energy into another, and this involves entropy (like the well-known Carnot limit). But to speak of efficiency for entropy does not make sense to me.

page 6, line 4: How can two unknowns (GEE and Reco) be estimated from one equation? I think there is some information missing here.

page 6, line 8: The authors convert the units from W m-2 K-1 to kJ m-2 K-1. The unit should be kJ m-2 K-1 month-1 (i.e., the time is missing, throughout the whole

manuscript), since entropy production refers to a rate, and not to an amount. But I do not understand the motivation for not keeping the units of W m-2 or W m-2 K-1 that are much more standard. Some explanation why this has been done would be helpful.

page 6, line 14: Radiative entropy production actually includes a factor of 4/3, as it does not deal with heat, but with radiation (the additional contribution of 1/3 is due to radiation pressure). I think it needs a brief explanation why this factor was omitted.

page 7, line 2: What do you mean by "to calculate the change in entropy of the metabolic system". Do you refer to entropy production? If you want to estimate entropy production, this would relate to dissipation of carbohydrates, which in turn relates to respiration. So I do not understand why NEE is being used.

page 7, line 14: Why is net longwave radiation being used to calculate entropy production? The entropy fluxes of longwave radiation are Rl,down/Tsky and Rl,up/Tsrf as the authors write earlier in the manuscript. But this is not the same as Rl,net * (1/Tsrf - 1/Tsky). (See major comment above)

page 7, line 20: dS/dt refers to the change in entropy with time, not change in entropy production. It should be zero in steady state, otherwise one cannot calculate entropy production from entropy fluxes. (See major comment above)

page 7, line 29/30: Why are these expressions referred to as MEP? I see no connection to MEP. They just formulate radiative entropy production. Also, what's the difference to Eq. 3.6 and 3.7?

page 8, line 3: "an ecosystem maximizes its entropy production when it converts all incoming Rs and Rl into work". This is not correct. First, work is something different than entropy production. Second, it is impossible to convert all incoming radiation into work, as it would imply that there is no energy left to maintain a temperature that is greater than T = 0K.

page 8, line 3: "... MEP.. is often negative or 0". No! Entropy production must always

be greater or equal to zero, otherwise there is something wrong in the formulations! Spontaneous reductions in entropy are only possible at the microscopic scale during extremely short time periods but are practically irrelevant at the scale of ecosystems.

page 8, line 7: "maximum entropy of metabolism". What do you mean by this?

page 8, line 13: You express the efficiency as the ratio of the entropy flux associated with net ecosystem exchange to the energy flux of GEE. Should this not compare gross energy fluxes, rather than net exchange to gross exchange?

page 8, line 16: This expression merely describes a radiative entropy flux, but not entropy production, or a maximum in entropy production.

page 8, line 18: This expression does not give an efficiency, because in steady state (a condition needed to estimate entropy production from fluxes), dS/dt = 0 so this expression is zero as well.

I stop here with commenting, because I think that the methodology has a number of flaws that I wonder how much these impact the results. In addition, as expressed earlier, I think that the overall motivation for this entropy-based analysis needs to be improved.

———————————————

---

## Referee Comment (RC2) · Anonymous Referee #2 · 28 Sep 2018

Overall summary

Wiesner et al. analyze energy use efficiencies for three different forested ecosystems along a moisture gradient using the framework of maximum entropy production. This is an interesting approach but needs a far more critical overview and careful analyses than what is presented currently.

Major Comments

The authors need to provide a more detailed overview of the concept of entropy. Are these concepts definable for biological systems? What are the caveats? How do they fit in with the second law of thermodynamics (and concepts of disorder and free energy)?

[Figure]

The framework presented in this study is built on Stoy et al., 2014, which in turn used a formulation by Holdway et al., 2010. These essentially simplify the concept of entropy to temperature normalization of fluxes of energy, carbon and water exchange. While a temperature normalized index for these quantities is likely to be highly useful in itself, does it warrant invoking entropy? Moreover, there are several inconsistencies, and not adequate explanation for how entropy for different fluxes is estimated. For instance, eq 4.6. which the authors define as the entropy efficiency of metabolism, is essentially a ratio of NEE:GPP. This has been previously identified as carbon use efficiency and extensively studied (for. e.g. see DeLucia et al., 2007 and references therein).

In many instances, it is unclear how energy and entropy are related. It would be useful to present side-by-side comparisons.

Three examples

1. Page 3, line 3: how does the entropy dissipation through sensible heat relate to energy dissipation? These concepts need to be clarified. 2. Fig. 4. Why look at JLE instead of LE fluxes? What is additionally learned from this? 3. Page 10, line 31. JNEE not being related to soil moisture. This claim (I say claim since data is not shown) would be highly interesting if it is contrasted with the NEE response to soil moisture. There are more rigorous formulations (e.g. Wu et al., 2017) as well as critical discussions (e.g. Volk and Paulus, 2010).

Another cause for concern is that that inferences are not quantitatively supposed. There are several instances where analysis is restricted to 'eyeballing' relationships between different curves, and correlation coefficients are not presented. In some occasions this leads to the authors making inferences that are not backed up by the data that is presented.

The writing is overly descriptive, and often disconnected with the conclusions. Is this study describing entropy fluxes and efficiency ratios and how these vary with different environmental conditions, or is it trying to use these variables to understand site dif-

ferences? The result is an unclear combination of the two. I would recommend the authors to stick to a storyline that is supported by the data.

Finally, there are several instances where the authors discuss the effect of soil moisture and rainfall on various fluxes/processes in the text (e.g. lines 13,19, 31 on page 10, line 25 on page 11) but do not choose to show these data. In my opinion these data are critical and need to be discussed (since it is a drought recovery study).

In light of these observations, I would not recommend this manuscript in its current form for publication in Biogeosciences. I think the authors provide very valuable observations, but should consider either re-framing the study or provide a more critical discussion on the concept of MEP, as well as consider extensive revisions on the writing as well as presentation of data.

Figures

There are several instances where curves are classified as significantly different, but do not appear significantly different from each other at all (Fig. 1d, for instance). The authors need to expand figure captions, since in the current form it is hard to infer what is being shown. E.g. Figure 4 has three time series (one for each site in most panels) but only one for sub panels b and e. It is unclear what data are presented. There are similar issues with Figs. 5-7.

I also feel that the authors rely on too much on summarizing data and do not explain how or why this is done (again, eg. Fig 4b and d). What are the data that are presented in these analyses?

The authors need to include sub panels in the text (Fig. 4a, b etc.).

Figure 1 has inconsistent units for temperature. For instance, subpanels c and e are plotted in units of Kelvin but d and f are in deg. C. Also, VPD is plotted in Figure 1 but not discussed at all amongst other discussions of Fig. 1 (Sec. 3.1).

Fig 2. Why are monthly means shown here, while the rest of the paper annual means

are presented?

Table 1: Please provide LAI estimates (if available) and also disturbance history, since this is a key component of your overall conclusions.

Minor comments

Line 1-2: Turbulent exchange of... specify (for e.g. momentum, heat, gases). Line 3: Maybe just use examples related to terrestrial ecosystems? Are these examples of the butterfly effect in terrestrial ecosystems?

Page 5 Lines 5-9: This assumes energy balance closure. Please describe why you closed the energy balance.

Eq. 2: Describe briefly how NEE was partitioned into source and sink terms.

eq. 3.6. and 3.7: Unclear why net fluxes are used. Line 23: Are periods of rainfall excluded from the analyses? Where is this described? eq 4.1 and 4.2: Why is 4.1. formulated using incoming radiation whereas as 4.2 using net fluxes?

eq. 4.8 is essentially carbon use efficiency (see major comment above).

Line 11. Subpanels missing. Lines 21-24: temperatures differences do not appear to be significantly different across sites in Fig. 1.

Page 10.

Sec. 3.2. Methods for this analysis are not presented. I think this section should be

merged with Sec. 2.1. (site description), as it doesn't appear to be a result of this study (unless methods are presented). Line 14: Soil moisture data seems important here (and in other places). Line 15: VPD effects are discussed first but EVI figure shown first in Fig. 4. Line 23: This is not correct according to Fig. 4. Line 23: See major comment above.

Page. 13

Line 1: What does 'preservation' on LE mean? Again, these are hard to interpret in the absence of absolute fluxes (see major comment above). Line 8: Ecosystems do not 'experience' LE (or JLE), but rather the interactions between the ecosystem and the overlying atmosphere determines the LE flux. Line 13: Clarify what this means.

Line 8: should read "at the more biodiverse site (i.e. mesic)" Line 11: What was the contribution of the C4 understory photosynthesis to overall ecosystem photosynthesis? Did you measure this? Lines 25-30: This is incorrect. Annual (and monthly) changes in EVI do not reflect changes in biomass. Biomass includes the carbon stored in the trunks, branches and stems of trees (among other pools), which do not fluctuate in forests at these timescales. Instead, at these timescales EVI is a measure of canopy greenness that is related to net photosynthesis (see Sims et al., 2008).

References

DeLucia, E.H., Drake, J.E., Thomas, R.B. and Gonzalez‐Meler, Miquel., 2007. Forest carbon use efficiency: is respiration a constant fraction of gross primary production? Global Change Biology, 13(6), pp.1157-1167.

Sims, D.A., Rahman, A.F., Cordova, V.D., El-Masri, B.Z., Baldocchi, D.D., Bolstad, P.V., Flanagan, L.B., Goldstein, A.H., Hollinger, D.Y., Misson, L. and Monson, R.K., 2008. A new model of gross primary productivity for North American ecosystems based solely on the enhanced vegetation index and land surface temperature from MODIS. Remote

Sensing of Environment, 112(4), pp.1633-1646.

Volk, T. and Pauluis, O., 2010. It is not the entropy you produce, rather, how you produce it. Philosophical Transactions of the Royal Society of London B: Biological Sciences, 365(1545), pp.1317-1322.

Wu, Z., Wu, X., Yang, Z. and Ouyang, L., 2017. A simple thermodynamic model for evaluating the ecological restoration effect on a manganese tailing wasteland. Ecological Modelling, 346, pp.20-29.

---

## Author Comment (AC1) · 14 Oct 2018

Title: Quantifying energy use efficiency via maximum entropy production: A case study from longleaf pine ecosystems Reviewer 1: Alex Kleidon Reviewer's comment: First, the entropy balance is used in Eq. 9, stating that the "overall change in entropy production (S) over time (t) in kJ m-2 K-1 of the ecosystem [is estimated] by adding entropy flux and entropy production". This is incorrect. What Eq. 9 formulates is the entropy balance. It balances the change in entropy on the left hand side of the equation (dS/dt) with the sum of all entropy exchange fluxes (J) and all entropy production terms ($\sigma$). This balance is typically assumed to be zero in a steady state, i.e., dS/dt = 0,

which then allows one to diagnose entropy production from the difference in entropy exchange fluxes. This is in fact what the authors do to diagnose entropy production in Eqs. 3.6 and 3.7 to diagnose entropy production by absorption of radiation. Yet, the authors later use dS/dt in Eq. 4.8 to derive an efficiency. This efficiency should be zero, otherwise they did not do the balancing correctly. So there is a major inconsistency in the methodology that needs to be resolved.

Authors' response: Thank you for your comment, you are correct, and acknowledge the wrong use of the term dS/dt. For the revisions we changed this calculation method and instead now focus on the entropy outputs and inputs and internal entropy production to quantify if dS/dt = 0 holds in our systems, which we have added to the revisions of the manuscript.

Reviewer's comment: Second, entropy production by absorption of longwave radiation is estimated using net longwave radiation at the surface (Eq. 3.7). What is the justification for using net long- wave radiation, rather than gross fluxes? After all, the downwelling longwave radiation of the surface adds an entropy flux of Rldown/Tsky, while the emission of radiation from the surface exports entropy at the rate of Rlup/Tsrf. Using the difference of these two fluxes (assuming that dS/dt=0) yields an entropy production of $\sigma$ = Rlup/Tsrf - Rl- down/Tsky, which is not the same as (Rlup - Rldown) * (1/Tsrf - 1/Tsky). The authors should correct this, or explain why their expression is justified. The same reasoning applies to the application of net ecosystem exchange, where I think that also gross fluxes should be used, not net fluxes.

Authors' response: Thank you for pointing out the mistake. We have adjusted our calculations following the Brunsell et al. (2011) approach using incoming longwave radiation as follows: Rl,in x (1/Tsrf-1/Tsky). We acknowledge that calculating the Rl,up/Tsrf and Rl,down/Tsky will estimate the incoming and outgoing entropy transfer associated with longwave radiation, but not the entropy produced due to absorption of longwave radiation and conversion to heat during this process (as shown in Brunsell et al. 2011). We have also changed our analysis to using half-hourly gross fluxes of GEE and Reco

following your comment.

Reviewer's comment: Additional insights gained from entropy fluxes and entropy production The authors link their entropy-based analysis to rather general concepts such as resilience and energy use efficiency. Yet, I do not see the additional insights gained by using entropy production, rather than an analysis based on the entropy, water, and carbon balance. Why does the entropy-based analysis provide more or novel insights that cannot be obtained by just an interpretation based on fluxes? The authors do not really answer this question within the manuscript and do not use the results to show this, as they only focus on an entropy-based analysis.

In terms of interpreting the observations, I think that there is a critical step missing that relates the observed differences to an interpretation of processes, and this cannot be gained by just looking at entropy. For instance, temperature changes result from changes in the energy balance, as temperature is a measure for heat content. Yet, the energy balance is not even shown or discussed. Likewise, to understand changes in evaporation, I would expect a water balance being discussed. Instead, this study directly diagnoses entropy fluxes and thereby skips this process-based level of interpretation. It does not show and interpret the fluxes of the energy, water, and carbon balances separately, and does not demonstrate that something else can be learned by looking at entropy. By lumping all aspects of the land surface into entropy production, I think that this neglects those aspects that are relevant for ecosystems from those that are irrelevant. The relevant flux for ecosystems is primarily the uptake of carbon, as this provides the chemical energy for terrestrial ecosystems. Plants live from the energy they fix during carbon assimilation, and, quite frankly, care little about the entropy production of other processes. For this manuscript to provide more solid insights, I think it needs a more process based interpretation using the available data, it needs to be more specific regarding those terms that are really relevant to ecosystems, and it needs to at least discuss why there is more to be gained by looking at entropy-based diagnostics.

Authors' response: Thank you, we have added an analysis and discussion of energy fluxes and the sites' energy balances to show the novelty of the entropy approach and to highlight that entropy production gives more insights about the energy efficiencies and resilience to drought at our sites.

We have included soil moisture content and rainfall in our analysis to quantify changes in entropy fluxes and entropy production due to changes in soil moisture and rainfall, but an analysis of the entire water budget was beyond the scope of this research project. We kindly disagree with the reviewers comment that the relevant flux for ecosystems is solely the carbon flux. For ecosystems (encompassing not only plant organisms), the partitioning of heat fluxes plays a significant role in their function, because the physical and biological processes are interconnected. LE in particular plays a large role in the maintenance of the surface temperature in ecosystems and is one of the largest contributors to entropy export in our ecosystem. However, we have adjusted our analysis according to your comment and are now using gross fluxes of GEE and Reco, to estimate the entropy change due to metabolic processes at the three sites. Minor comments:

Abstract: "Our study provides foundational evidence of how MEP can be used to determine resiliency across ecosystems globally" - I am not at all convinced and doubt this conclusion. The authors provide no discussion why a diagnosis based on entropy fluxes yields more or better insights than the diagnosis of energy, water, and carbon balances. I see this as a critical missing bit in this manuscript.

Authors' response: We will adjust the discussion and methodology accordingly, to show that entropy production indeed gives more insights about differences in resilience at the three longleaf pine sites, in contrast to solely using energy fluxes. In our revisions we will focus more on the entropy import and export, as well as the internal entropy production, to quantify how close these systems are to a thermodynamic steady state. This could not be accomplished using solely energy fluxes.

Introduction, page 2, line 16: MEP is referred to as a principle in the text. At best, it is a "proposed" principle, or better hypothesis, as it is not generally being accepted. Authors' response: Thank you for your comment. We will adjust the sentence accordingly. page 2, line 24: How can agricultural systems exceed MEP if MEP already describes the maximum? This does not make sense. What I can imagine is that agricultural systems maintain a different state because of nutrient inputs, but then, the boundary conditions are changed because there are additional exchange fluxes across the system boundary. Also, why would this excessive entropy production be unsustainable? As long as the nutrient input can be maintained, I see no reason why it should be unsustainable.

Authors' response: In the papers we cited (Patzek et al. 2008 and Steinborn and Svirezhev 2000) the energy efficiency of agricultural management practices was determined, amongst other things using entropy metrics. Patzek et al. determined this unsustainability using gross and net primary productivity measures, as well as biomass estimates. The change in entropy was determined by quantifying anthropogenic energy inputs (fertilizers, herbicides, pesticides, fossil fuels, electricity, etc.) and energy production and respiration rates by the crops, and comparing it to a system which the agricultural practice displaced (here a prairie system). So here the "maximum entropy production" was equal to the productivity of an unmanaged ecosystem. The authors note that: "Excess entropy generated in an agrosystem manifests itself mostly as soil degradation by chemical and mechanical means, and toxic effluent runoff." This basically implies that intensive agriculture requires energy from "outside the boundaries" (i.e. fertilizer production elsewhere) to meet increasing production demands and to balance soil degrading processes, due to the intensive management. Steinborn and Svirezhev (2000) provide similar measures, showing that a decrease in energy inputs from anthropogenic sources, or an increase in biomass production at similar energy inputs could decrease the excess of entropy production and therefore make the system more sustainable. We expand on this in the revisions of the paper as well.

page 3, line 9: What are entropy efficiency ratios? In thermodynamics, efficiency is used to describe the conversion efficiency of one form of energy into another, and this involves entropy (like the well-known Carnot limit). But to speak of efficiency for entropy does not make sense to me.

Authors' response: Thank you for your comment, we refer to these ratios as entropy efficiency ratios, as what they are describing in relation to maximum entropy production are how close these systems are to MEP. In the revised version we describe this more effectively to eliminate possible confusion. However, following your comment we have changed the section concerned with the ratio of all ecosystem fluxes. We are now quantifying how close these systems are to a steady state by estimating $dS/dt$ using entropy inputs and outputs and internal entropy production. The revised version will solely quantify the ratio of internal entropy production of radiation to MEP.

page 6, line 4: How can two unknowns (GEE and Reco) be estimated from one equation? I think there is some information missing here.

Authors' response: Thank you for your comment, this method was described in more detail in other published studies from this lab following common approaches to partitioning eddy covariance data. We now add a more detailed description of how these estimates are obtained.

page 6, line 8: The authors convert the units from W m-2 K-1 to kJ m-2 K-1. The unit should be kJ m-2 K-1 month-1 (i.e., the time is missing, throughout the whole manuscript), since entropy production refers to a rate, and not to an amount. But I do not understand the motivation for not keeping the units

Authors' response: We have adjusted the units accordingly. page 6, line 14: Radiative entropy production actually includes a factor of 4/3, as it does not deal with heat, but with radiation (the additional contribution of 1/3 is due to radiation pressure). I think it needs a brief explanation why this factor was omitted. Authors' response: We avoided this factor previously, due to the controversy surrounding this factor (see Ozawa et al.

2003; Kleidon and Lorenz, 2005; Fraedrich and Lunkeit, 2008; Kleidon, 2009; Pascale et al., 2012) and because we assumed that the incoming and outgoing radiation does not assert radiation pressure. We will add more explanation of why we chose to omit this factor to the revisions.

page 7, line 2: What do you mean by "to calculate the change in entropy of the metabolic system". Do you refer to entropy production? If you want to estimate entropy production, this would relate to dissipation of carbohydrates, which in turn relates to respiration. So I do not understand why NEE is being used. Authors' response: For the revised paper we have adjusted our analysis accordingly and are now estimating the change in entropy due to metabolic processes using the half-hourly gross fluxes of Reco and GEE.

page 7, line 14: Why is net longwave radiation being used to calculate entropy production? The entropy fluxes of longwave radiation are Rl,down/Tsky and Rl,up/Tsrf as the authors write earlier in the manuscript. But this is not the same as Rl,net * (1/Tsrf - 1/Tsky). (See major comment above)

Authors' response: Please see our response above. page 7, line 20: dS/dt refers to the change in entropy with time, not change in entropy production. It should be zero in steady state, otherwise one cannot calculate entropy production from entropy fluxes. (See major comment above) Authors' response: You are correct, and we acknowledge the wrong use of dS/dt. As noted above, for the revisions we will change this calculation method. page 7, line 29/30: Why are these expressions referred to as MEP? I see no connection to MEP. They just formulate radiative entropy production. Also, what's the difference to Eq. 3.6 and 3.7?

Authors' response: Thank you for your comment, as we note following Eq. 4.3 under an ideal case MEPRL would be zero, if the system transfers all energy from available radiation into LE, rather than H. To avoid confusion, we will revise this section and make it clearer that we are talking about an assumption or an empirical maximum

entropy production (as shown in Stoy et al. 2014). Even if this assumption does not necessarily reflect reality, it still gives us a means to compare different ecosystems or sites with each other with respect to how they reflect, absorb and emit radiation.

page 8, line 3: "an ecosystem maximizes its entropy production when it converts all incoming Rs and Rl into work". This is not correct. First, work is something different than entropy production. Second, it is impossible to convert all incoming radiation into work, as it would imply that there is no energy left to maintain a temperature that is greater than T = 0K.

Authors' response: As noted above, we will change the dS/dt section, which will exclude this analysis. Instead we will quantify the sum of entropy imports and exports, as well as entropy production, to determine dS/dt.

page 8, line 3: ". . . MEP.. is often negative or 0". No! Entropy production must always be greater or equal to zero, otherwise there is something wrong in the formulations! Spontaneous reductions in entropy are only possible at the microscopic scale during extremely short time periods but are practically irrelevant at the scale of ecosystems. Authors' response: We note that this sentence refers to the empirical maximum entropy production of shortwave and longwave radiation. MEP of longwave radiation is usually small when considered as part of the whole system. Here we assume that an ideal system partitions all incoming energy into LE (and M and G), rather than H, such that the temperature difference between surface temperature and the temperature of the overlying air mass would approach zero ($T_{srf} = T_{air}$, ref. Stoy et al. 2014). We have added a better description of that assumption to the methods.

page 8, line 7: "maximum entropy of metabolism". What do you mean by this? Authors' response: We indeed intended to refer to the maximum decrease of entropy due to C assimilation of plant organisms in our systems, calculated by quantifying energy uptake as Ein from GEE and the simultaneous decrease in entropy. We have changed our analysis to using gross fluxes of GEE and Reco and we are now calculating the

change in entropy of metabolic processes from the time of day when these fluxes occur. We will elaborate on this more in the revised document.

page 8, line 13: You express the efficiency as the ratio of the entropy flux associated with net ecosystem exchange to the energy flux of GEE. Should this not compare gross energy fluxes, rather than net exchange to gross exchange.

Authors' response: In the revised paper we are now estimating the entropy decrease due to C assimilation during the day and the entropy increase through Reco during day and night, to calculate the change in entropy of metabolic processes. page 8, line 16: This expression merely describes a radiative entropy flux, but not entropy production, or a maximum in entropy production.

Authors' response: Thank you for your comment. As we altered the section including the whole ecosystem entropy budget, this section was omitted.

page 8, line 18: This expression does not give an efficiency, because in steady state (a condition needed to estimate entropy production from fluxes), dS/dt = 0 so this expression is zero as well.

Authors' response: As noted above, we are altering the calculation method to describe ecosystem efficiency in terms of entropy fluxes and production by focusing on the ratio of entropy outputs to inputs, as well as the internal production of entropy. I stop here with commenting, because I think that the methodology has a number of flaws that I wonder how much these impact the results. In addition, as expressed earlier, I think that the overall motivation for this entropy-based analysis needs to be improved. Authors' response: We will clarify the motivation of this study, as it was not as transparent as we intended. With this study we wanted to show that sites exhibit differences in their energy use efficiencies due to differences in energy partitioning and entropy production, which in part is due to differences in surface, air and sky temperatures. The Intermediate site for example maintained higher surface and air temperatures compared to the other sites (except for the years 2012 and 2013), which for example lowered its

entropy production and the entropy flux of LE.

---

## Author Comment (AC2) · 14 Oct 2018

Response to reviewer 2 Major Comments Reviewer's comment: The authors need to provide a more detailed overview of the concept of entropy. Are these concepts definable for biological systems? What are the caveats? How do they fit in with the second law of thermodynamics (and concepts of disorder and free energy)?

Authors' response: Thank you for this comment. We will add more background information on the concept of entropy, specifically tailored towards biological systems, with our revisions.

[Figure]

Reviewer's comment: The framework presented in this study is built on Stoy et al., 2014, which in turn used a formulation by Holdway et al., 2010. These essentially simplify the concept of entropy to temperature normalization of fluxes of energy, carbon and water exchange. While a temperature normalized index for these quantities is likely to be highly useful in itself, does it warrant invoking entropy? Moreover, there are several inconsistencies, and not adequate explanation for how entropy for different fluxes is estimated. For instance, eq 4.6. which the authors define as the entropy efficiency of metabolism, is essentially a ratio of NEE:GPP. This has been previously identified as carbon use efficiency and extensively studied (for. e.g. see DeLucia et al., 2007 and references therein). In many instances, it is unclear how energy and entropy are related. It would be useful to present side-by-side comparisons.

Author's response: You are correct. The concept of entropy applied in our study is essentially a normalization of energy fluxes to temperature. We are now using half-hourly gross fluxes of GEE and Reco to quantify the change in entropy of metabolic processes. As these fluxes occur at different temperatures (GEE during the day and Reco during day and night); this will go beyond an analysis of the carbon use efficiency.

Three examples Reviewer's comment: 1. Page 3, line 3: how does the entropy dissipation through sensible heat relate to energy dissipation? These concepts need to be clarified. 2. Fig. 4. Why look at JLE instead of LE fluxes? What is additionally learned from this? 3. Page 10, line 31. JNEE not being related to soil moisture. This claim (I say claim since data is not shown) would be highly interesting if it is contrasted with the NEE response to soil moisture. There are more rigorous formulations (e.g. Wu et al., 2017) as well as critical discussions (e.g. Volk and Paulus, 2010).

Authors' response: An analysis of entropy fluxes is preferable in ecosystems which are exposed to different environmental variables, as differences in surface and air temperatures can affect entropy production of energy fluxes of LE and H. For example, two systems could have similar magnitudes of LE, but differ in JLE due to differences in air or surface temperatures. For the ecosystem which maintains a higher surface/air

temperature, the entropy flux would be lower, suggesting that it is less efficient in exporting entropy across its boundaries. By calculating the difference between entropy outputs and inputs, as well as internal entropy production, one can estimate how close an ecosystem is to a thermodynamic "steady state" and therefore how organized it is. This could not be accomplished by looking only at the energy balance. We will add a more thorough introduction and discussion of why entropy metrics can be more useful in describing energy use efficiency. In our revised manuscript we will include figures for all results. Reviewer's comment: Another cause for concern is that that inferences are not quantitatively supposed. There are several instances where analysis is restricted to 'eyeballing' relationships between different curves, and correlation coefficients are not presented. In some occasions this leads to the authors making inferences that are not backed up by the data that is presented. Author's response: We will add tables of Type III effect summaries for all models as supplementary tables, as well as add indicators to the plots where differences between factors were significantly different.

Reviewer's comment: The writing is overly descriptive, and often disconnected with the conclusions. Is this study describing entropy fluxes and efficiency ratios and how these vary with different environmental conditions, or is it trying to use these variables to understand site differences? The result is an unclear combination of the two. I would recommend the authors to stick to a storyline that is supported by the data.

Authors' response: Thank you for this valuable comment. We included the effects of environmental variables to understand changes and differences in entropy production and fluxes and thus changes in energy efficiency at our three sites. In our revisions we will make this more clear.

Reviewer's comment: Finally, there are several instances where the authors discuss the effect of soil moisture and rainfall on various fluxes/processes in the text (e.g. lines 13,19, 31 on page 10, line 25 on page 11) but do not choose to show these data. In my opinion these data are critical and need to be discussed (since it is a drought recovery study). Authors' response: You are correct, and we will add figures showing their

effects, as we did include these variables in our models. However, rainfall often had no significant effect on the response variables by site or as a simple effect. Reviewer's comment: In light of these observations, I would not recommend this manuscript in its current form for publication in Biogeosciences. I think the authors provide very valuable observations, but should consider either re-framing the study or provide a more critical discussion on the concept of MEP, as well as consider extensive revisions on the writing as well as presentation of data.

Authors' response: Thank you. We will add a more thorough discussion. Figures

Reviewer's comment: There are several instances where curves are classified as significantly different, but do not appear significantly different from each other at all (Fig. 1d, for instance). The authors need to expand figure captions, since in the current form it is hard to infer what is being shown. E.g. Figure 4 has three time series (one for each site in most panels) but only one for sub panels b and e. It is unclear what data are presented. There are similar issues with Figs. 5-7.

Authors' response: We will add a supplementary file with all type-3 tests of effects for all models included in this manuscript to show significant differences among the independent variables where applicable. For Figure 4, 5 and 7 we kindly refer to the Figure captions, where we note that when only one black line is shown, the interaction with site was not significant. For example, for Figure 4, we wrote: "For (b) and (e) the interaction with site was not significant."

Reviewer's comment: I also feel that the authors rely on too much on summarizing data and do not explain how or why this is done (again, eg. Fig 4b and d). What are the data that are presented in these analyses?

Authors' response: Following your suggestion, we are changing our analysis to estimate entropy from mean half hourly energy fluxes for daily time-steps (W m-2 K-1) to look at differences in energy and entropy metrics.

[Figure]

Reviewer's comment: The authors need to include sub panels in the text (Fig. 4a, b etc.).

Authors' response: Thank you for this suggestion. For our revisions we are adding sub panel information to the text.

Reviewer's comment: Figure 1 has inconsistent units for temperature. For instance, subpanels c and e are plotted in units of Kelvin but d and f are in deg. C. Also, VPD is plotted in Figure 1 but not discussed at all amongst other discussions of Fig. 1 (Sec. 3.1).

Authors' response: We used air temperature in degrees C for our model formulations, but for the calculation of entropy we converted air temperature to Kelvin. We will change Figure 1 to have consistent units. We will also add a more thorough discussion for VPD.

Reviewer's comment: Fig 2. Why are monthly means shown here, while the rest of the paper annual means are presented?

Authors' response: We used monthly means of entropy transfers and production for all of our analyses. As we are now using mean daily estimates, we will revise this in the text.

Reviewer's comment: Table 1: Please provide LAI estimates (if available) and also disturbance history, since this is a key component of your overall conclusions.

Authors' response: We are adding LAI data when available (for the mesic and xeric site), as well as disturbances history for all sites to the Table.

Minor comments Reviewer's comment: Page 2 Line 1-2: Turbulent exchange of... specify (for e.g. momentum, heat, gases). Line 3: Maybe just use examples related to terrestrial ecosystems? Are these examples of the butterfly effect in terrestrial ecosystems?

Authors' response: Thank you for your comment. We have added heat fluxes to the

sentence, as part of our story was focusing on the partitioning and entropy transfer through LE and H. We will also add examples of the butterfly effect in ecosystems.

Reviewer's comment: Page 5 Lines 5-9: This assumes energy balance closure. Please describe why you closed the energy balance.

Authors' response: To accurately describe the entropy balance for ecosystems, we are required to have a closed energy balance. Following this and the comments from reviewer 1 (Alex Kleidon) we will add a more detailed description of the energy balance and energy partitioning at the three sites, which will include a better description of why we chose to close the energy balance using the Bowen method.

Reviewer's comment: Page 6 Eq. 2: Describe briefly how NEE was partitioned into source and sink terms.

Authors' response: Thank you for your comment. We will add a more detailed description on how we partitioned NEE into GEE and Reco following Whelan et al. (2013) and Starr et al. (2016).

Reviewer's comment: Page 7 eq. 3.6. and 3.7: Unclear why net fluxes are used. Line 23: Are periods of rainfall excluded from the analyses? Where is this described? eq 4.1 and 4.2: Why is 4.1. formulated using incoming radiation whereas as 4.2 using net fluxes? Authors' response: We have revised this section and are estimating entropy production as follows:

and

Reviewer's comment: Page 8 eq. 4.8 is essentially carbon use efficiency (see major comment above).

Authors' response: We are revising our analysis accordingly, and we are now using half-hourly gross fluxes of GEE and Reco to quantify the change in entropy of metabolic processes. As these fluxes occur at different temperatures (GEE during the day and Reco during day and night), this will go beyond an analysis of the carbon use efficiency.

Reviewer's comment: Page 9 Line 11. Subpanels missing. Lines 21-24: temperatures differences do not appear to be significantly different across sites in Fig. 1. Authors' response: We kindly note that even though these figures appear to not show significant differences in temperature across the sites, our statistical results indicate that there were in fact significant differences in temperature. We are adding the type-3 tests for all models to show this.

Reviewer's comment: Page 10. Sec. 3.2. Methods for this analysis are not presented. I think this section should be merged with Sec. 2.1. (site description), as it doesn't appear to be a result of this study (unless methods are presented). Line 14: Soil moisture data seems important here (and in other places). Line 15: VPD effects are discussed first but EVI figure shown first in Fig. 4. Line 23: This is not correct according to Fig. 4. Line 23: See major comment above.

Authors' response: Thank you for your comment. We have added a description of the methods that were used to calculate understory biomass for the three sites. We are also adding graphs for the independent variable of SWC. We are changing the description of VPD and EVI according to the order shown in the figures and will make sure that this is consistent throughout the text as well.

Reviewer's comment: Page. 13 Line 1: What does 'preservation' on LE mean? Again, these are hard to interpret in the absence of absolute fluxes (see major comment above). Line 8: Ecosystems do not 'experience' LE (or JLE), but rather the interactions between the ecosystem and the overlying atmosphere determines the LE flux. Line 13: Clarify what this means.

Authors' response: Thank you for your comment. As noted above, we are adding an analysis of the energy balance to show absolute fluxes. We will make sure to not "personalize" ecosystems throughout this manuscript.

Reviewer's comment: Page 14 Line 8: should read "at the more biodiverse site (i.e. mesic)" Line 11: What was the contribution of the C4 understory photosynthesis to

overall ecosystem photosynthesis? Did you measure this? Lines 25-30: This is incorrect. Annual (and monthly) changes in EVI do not reflect changes in biomass. Biomass includes the carbon stored in the trunks, branches and stems of trees (among other pools), which do not fluctuate in forests at these timescales. Instead, at these timescales EVI is a measure of canopy greenness that is related to net photosynthesis (see Sims et al., 2008). Authors' response: We have changed the sentence on line 8 accordingly. Unfortunately, we did not measure differences of the C4 understory at our sites over the course of this study. However, in Wiesner et al. (2018) we showed that the understory contributes about 50% to Reco, using soil respiration data. We will correct the definition of EVI. What EVI reflects is indeed the change in LAI (canopy

---

## Author Response (AR1)

College of Arts and Sciences

Department of Biological Sciences

[Figure]

Dr. Gregory Starr, Professor                           gstarr@ua.edu
Department of Biological Sciences                       205-348-0556
3097A Shelby Hall
Tuscaloosa, AL. 35487

December 15, 2018

Dear Dr. Still

Please find enclosed our revised manuscript, mss# bg-2018-322, title, "Quantifying energy use efficiency via entropy production: A case study from longleaf pine ecosystems." We have completed all revisions per the Reviewer's' comments and suggestion. We would like to thank them for their constructive reviews which have help to improve the manuscript. Below are the Reviewer's' comments in normal black font and our responses to the comments in blue. We have include in our online submission both a marked and clean copy of the revised manuscript. Please contact me if you have any questions regarding the manuscript.

Sincerely,

Gregory Starr, Ph.D.
Professor of Global Change Biology
University of Alabama

**Reviewer 1: Alex Kleidon**

**Reviewer's comment:** *First, the entropy balance is used in Eq. 9, stating that the "overall change in entropy production (S) over time (t) in kJ m-2 K-1 of the ecosystem [is estimated] by adding entropy flux and entropy production". This is incorrect. What Eq. 9 formulates is the entropy balance. It balances the change in entropy on the left hand side of the equation (dS/dt) with the sum of all entropy exchange fluxes (J) and all entropy production terms (σ). This balance is typically assumed to be zero in a steady state, i.e., dS/dt = 0, which then allows one to diagnose entropy production from the difference in entropy exchange fluxes. This is in fact what the authors do to diagnose entropy production in Eqs. 3.6 and 3.7 to diagnose entropy production by absorption of radiation. Yet, the authors later use dS/dt in Eq. 4.8 to derive an efficiency. This efficiency should be zero, otherwise they did not do the balancing correctly. So there is a major inconsistency in the methodology that needs to be resolved.*

**Authors' response:** We have changed the calculation for dS/dt and now focused on the entropy outputs and inputs and internal entropy production, to quantify the change in entropy (dS/dt). Please see section 2.5 (Eq. 4.9) and the results section 3.6 in the revisions.

**Reviewer's comment:** *Second, entropy production by absorption of longwave radiation is estimated using net longwave radiation at the surface (Eq. 3.7). What is the justification for using net long- wave radiation, rather than gross fluxes? After all, the downwelling longwave radiation of the surface adds an entropy flux of Rldown/Tsky, while the emission of radiation from the surface exports entropy at the rate of Rlup/Tsrf. Using the difference of these two fluxes (assuming that dS/dt=0) yields an entropy production of σ = Rlup/Tsrf - Rl- down/Tsky, which is not the same as (Rlup - Rldown) \* (1/Tsrf - 1/Tsky). The authors should correct this, or explain why their expression is justified. The same reasoning applies to the application of net ecosystem exchange, where I think that also gross fluxes should be used, not net fluxes.*

**Authors' response:** Thank you for pointing out the mistake. We have adjusted our calculations following the Brunsell et al. (2011) approach using incoming longwave radiation to calculate entropy production as follows: $R_{l,in}$ x $(1/T_{srf}-1/T_{sky})$. We acknowledge that calculating the $R_{l,up}/T_{srf}$ and $R_{l,down}/T_{sky}$ will estimate the incoming and outgoing entropy transfer associated with longwave radiation, but not the entropy produced due to absorption of longwave radiation and conversion to heat during this process (as shown in Brunsell et al. 2011). Please see section 2.5 (Eq. 4.7) in the manuscript.

We have also eliminated the efficiency ratio calculations of metabolic activity and now quantify the overall change in metabolic energy using solely NEE. We believe that using direct fluxes is superior, as it avoids any influence of model bias, as $R_{eco}$ is estimated using temperature data. However, we have changed the analysis to comparing metabolic energy and entropy changes in the systems (see section 2.5 and the results section 3.5), rather than using a ratio to quantify metabolic efficiency. We have also changed our analysis to using daily average half-hourly fluxes for all variables in the manuscript following your comment.

**Reviewer's comment:** *Additional insights gained from entropy fluxes and entropy production The authors link their entropy-based analysis to rather general concepts such as resilience and energy use efficiency. Yet, I do not see the additional insights gained by using entropy production, rather than an analysis based on the entropy, water, and carbon balance. Why does the entropy-based analysis provide more or novel insights that cannot be obtained by just an interpretation based on fluxes? The authors do not really answer this question within the manuscript and do not use the results to show this, as they only focus on an entropy-based analysis.*

*In terms of interpreting the observations, I think that there is a critical step missing that relates the observed differences to an interpretation of processes, and this cannot be gained by just looking at entropy. For instance, temperature changes result from changes in the energy balance, as temperature is a measure for heat content. Yet, the energy balance is not even shown or discussed. Likewise, to understand changes in evaporation, I would expect a water balance being discussed. Instead, this study directly diagnoses entropy fluxes and thereby skips this process-based level of interpretation. It does not show and interpret the fluxes of the energy, water, and carbon balances separately, and does not demonstrate that something else can be learned by looking at entropy.*

*By lumping all aspects of the land surface into entropy production, I think that this neglects those aspects that are relevant for ecosystems from those that are irrelevant. The relevant flux for ecosystems is primarily the uptake of carbon, as this provides the chemical energy for terrestrial ecosystems. Plants live from the energy they fix during carbon assimilation, and, quite frankly, care little about the entropy production of other processes.*

*For this manuscript to provide more solid insights, I think it needs a more process based interpretation using the available data, it needs to be more specific regarding those terms that are really relevant to ecosystems, and it needs to at least discuss why there is more to be gained by looking at entropy-based diagnostics.*

**Authors' response:** We have added an analysis and discussion of energy fluxes and the sites' energy balances to show the novelty of the entropy approach and to highlight specifically that the inclusion of entropy production gives more insights about the energy efficiencies and ecosystem function. Please see sections 3.3 and 3.4 for the results. To estimate the entropy budget of ecosystems, it is of particular importance to quantify entropy production based on the absorption of radiation, as this term is of similar magnitude as the entropy fluxes of LE and H together at our sites. We have also added a more thorough introduction and discussion of the topic.

We have also included soil moisture content and rainfall in our analysis to quantify changes in entropy fluxes and entropy production, but an analysis of the whole water budget was beyond the scope of this research project.

We kindly disagree with the reviewer's comment that the relevant flux for ecosystems is solely the carbon flux. For ecosystems (encompassing not only plant organisms), the partitioning of heat fluxes plays a significant role in their function, because the physical and biological processes are interconnected. LE in particular plays a large role in the maintenance of the surface temperature in ecosystems and is one of the largest contributors to entropy export in our ecosystem.

**Minor comments:**

*Abstract: "Our study provides foundational evidence of how MEP can be used to determine resiliency across ecosystems globally" - I am not at all convinced and doubt this conclusion. The authors provide no discussion why a diagnosis based on entropy fluxes yields more or better insights than the diagnosis of energy, water, and carbon balances. I see this as a critical missing bit in this manuscript.*

**Authors' response:** We have adjusted the discussion and methodology to show that entropy metrics can give further insights about differences in ecosystem function at the three longleaf pine sites, in addition to using energy fluxes. Our revisions focus on the entropy import and export, as well as the internal entropy production, to quantify how close these ecosystems are to a thermodynamic steady state.

*Introduction, page 2, line 16: MEP is referred to as a principle in the text. At best, it is a "proposed" principle, or better hypothesis, as it is not generally being accepted.*

**Authors' response:** We have adjusted the sentence accordingly.

*page 2, line 24: How can agricultural systems exceed MEP if MEP already describes the maximum? This does not make sense. What I can imagine is that agricultural systems maintain a different state because of nutrient inputs, but then, the boundary conditions are changed because there are additional exchange fluxes across the system boundary. Also, why would this excessive entropy production be unsustainable? As long as the nutrient input can be maintained, I see no reason why it should be unsustainable.*

**Authors' response:** We have altered our introduction to focus more on the importance of entropy exchanges and entropy production in ecosystems. The section of MEP and MEP in agricultural systems has therefore been eliminated.

*page 3, line 9: What are entropy efficiency ratios? In thermodynamics, efficiency is used to describe the conversion efficiency of one form of energy into another, and this involves entropy (like the well-known Carnot limit). But to speak of efficiency for entropy does not make sense to me.*

**Authors' response:** We have adjusted our revisions to avoid the use of "entropy efficiency".

*page 6, line 4: How can two unknowns (GEE and Reco) be estimated from one equation? I think there is some information missing here.*

**Authors' response:** We have added a more detailed description of how these fluxes were obtained (see section 2.4).

*page 6, line 8: The authors convert the units from W m-2 K-1 to kJ m-2 K-1. The unit should be kJ m-2 K-1 month-1 (i.e., the time is missing, throughout the whole manuscript), since entropy production refers to a rate, and not to an amount. But I do not understand the motivation for not keeping the units*

**Authors' response:** We have adjusted the units accordingly and are now using daily averages of half-hourly energy and entropy fluxes in W $m^{-2}$ and W $m^{-2}$ $K^{-1}$, respectively.

*page 6, line 14: Radiative entropy production actually includes a factor of 4/3, as it does not deal with heat, but with radiation (the additional contribution of 1/3 is due to radiation pressure). I think it needs a brief explanation why this factor was omitted.*

**Authors' response:** We avoided this factor assuming that the incoming and outgoing radiation does not assert radiation pressure (see Ozawa et al. 2003; Kleidon and Lorenz, 2005; Fraedrich and Lunkeit, 2008; Kleidon, 2009; Pascale et al., 2012). Please see section 2.5.

*page 7, line 2: What do you mean by "to calculate the change in entropy of the metabolic system". Do you refer to entropy production? If you want to estimate entropy production, this would relate to dissipation of carbohydrates, which in turn relates to respiration. So I do not understand why NEE is being used.*

**Authors' response:** As noted above, we have changed the calculation of metabolic energy and entropy solely using NEE data without quantifying an efficiency ratio. Instead we are now comparing results for NEE energy and entropy.

*page 7, line 14: Why is net longwave radiation being used to calculate entropy production? The entropy fluxes of longwave radiation are Rl,down/Tsky and Rl,up/Tsrf as the authors write earlier in the manuscript. But this is not the same as Rl,net * (1/Tsrf - 1/Tsky). (See major comment above)*

**Authors' response:** We have altered this calculation method in section 2.5. Please see our response above.

*page 7, line 20: dS/dt refers to the change in entropy with time, not change in entropy production. It should be zero in steady state, otherwise one cannot calculate entropy production from entropy fluxes. (See major comment above)*

**Authors' response:** You are correct. As noted above, we have fixed this error.

*page 7, line 29/30: Why are these expressions referred to as MEP? I see no connection to MEP. They just formulate radiative entropy production. Also, what's the difference to Eq. 3.6 and 3.7?*

**Authors' response:** We have revised this section to make it clearer that we are talking about an assumption. If this assumption does not necessarily reflect reality, it still gives us a means to compare different ecosystems or sites with respect to how they reflect, absorb and emit radiation.

*page 8, line 3: "an ecosystem maximizes its entropy production when it converts all incoming Rs and Rl into work". This is not correct. First, work is something different than entropy production. Second, it is impossible to convert all incoming radiation into work, as it would imply that there is no energy left to maintain a temperature that is greater than T = 0K.*

**Authors' response:** As noted above, we have changed the dS/dt section, and have excluded this ratio analysis.

*page 8, line 3: ". . . MEP.. is often negative or 0". No! Entropy production must always be greater or equal to zero, otherwise there is something wrong in the formulations! Spontaneous reductions in entropy are only possible at the microscopic scale during extremely short time periods but are practically irrelevant at the scale of ecosystems.*

**Authors' response:** We have adjusted the sentence accordingly in section 2.6.

*page 8, line 7: "maximum entropy of metabolism". What do you mean by this?*

**Authors' response:** We apologize for the confusion with this statement. We have changed our analysis to using net fluxes and are now looking at metabolic activity, rather than efficiency at the site by comparing energy and entropy fluxes of NEE at the sites.

*page 8, line 13: You express the efficiency as the ratio of the entropy flux associated with net ecosystem exchange to the energy flux of GEE. Should this not compare gross energy fluxes, rather than net exchange to gross exchange*

**Authors' response:** We have excluded the analysis of metabolic efficiency in the revised paper.

*page 8, line 16: This expression merely describes a radiative entropy flux, but not entropy production, or a maximum in entropy production.*

**Authors' response:** We have altered the section including the whole ecosystem entropy budget; this section was omitted.

*page 8, line 18: This expression does not give an efficiency, because in steady state (a condition needed to estimate entropy production from fluxes), dS/dt = 0 so this expression is zero as well.*

**Authors' response:** As noted above, we have altered the calculation.

*I stop here with commenting, because I think that the methodology has a number of flaws that I wonder how much these impact the results. In addition, as expressed earlier, I think that the overall motivation for this entropy-based analysis needs to be improved.*

**Authors' response:** We have substantially altered the introduction and discussion in the revised manuscript to improve clarity about the methodology and as to why entropy metrics are useful in quantifying differences in ecosystem function.

**Response to reviewer 2**

**Major Comments**

**Reviewer's comment:** *The authors need to provide a more detailed overview of the concept of entropy. Are these concepts definable for biological systems? What are the caveats? How do they fit in with the second law of thermodynamics (and concepts of disorder and free energy)?*

**Author's response:** We have added more background information on the concept of entropy, specifically tailored towards biological systems.

**Reviewer's comment:** *The framework presented in this study is built on Stoy et al., 2014, which in turn used a formulation by Holdway et al., 2010. These essentially simplify the concept of entropy to temperature normalization of fluxes of energy, carbon and water exchange. While a temperature normalized index for these quantities is likely to be highly useful in itself, does it warrant invoking entropy? Moreover, there are several inconsistencies, and not adequate explanation for how entropy for different fluxes is estimated. For instance, eq 4.6. which the authors define as the entropy efficiency of metabolism, is essentially a ratio of NEE:GPP. This has been previously identified as carbon use efficiency and extensively studied (for. e.g. see DeLucia et al., 2007 and references therein). In many instances, it is unclear how energy and entropy are related. It would be useful to present side-by-side comparisons.*

**Author's response:** You are correct; the concept of thermodynamic entropy applied in our study is essentially a normalization of energy fluxes to temperature, as the magnitude of entropy fluxes and entropy production is a function of the temperature from which flux originated. This can be helpful in determining differences in energy use efficiency in ecosystems, specifically for the sites as these differed in sky, air, surface and soil temperatures. We are now using half-hourly fluxes for all energy and entropy in our calculations. Furthermore, we have omitted the section on metabolic efficiency ratios and are now focusing on metabolic activity in terms of energy and entropy (see section 2.5 and 3.5).

**Three examples**

**Reviewer's comment:** *1. Page 3, line 3: how does the entropy dissipation through sensible heat relate to energy dissipation? These concepts need to be clarified. 2. Fig. 4. Why look at JLE instead of LE fluxes? What is additionally learned from this? 3. Page 10, line 31. JNEE not being related to soil moisture. This claim (I say claim since data is not shown) would be highly interesting if it is contrasted with the NEE response to soil moisture. There are more rigorous formulations (e.g. Wu et al., 2017) as well as critical discussions (e.g. Volk and Paulus, 2010).*

**Author's response:** An analysis of entropy fluxes is preferable in ecosystems which are exposed to different environmental variables, such as differences in surface and air temperatures, which affect the magnitude of entropy fluxes and entropy production. For example, two systems could have similar magnitudes of LE, but differ in $J_{LE}$ due to differences in air or surface temperatures. For the ecosystem which maintains a higher surface/air temperature, the entropy flux would be lower, suggesting that it is less efficient in exporting entropy across its boundaries. By calculating the difference between entropy outputs and inputs, as well as internal entropy production, one can estimate how close an ecosystem is to a thermodynamic "steady state" and

therefore how organized it is. This cannot be accomplished by studying the energy balance alone. We have added a more thorough introduction and discussion of why entropy metrics can be more useful in describing energy use efficiency.

**Reviewer's comment:** *Another cause for concern is that that inferences are not quantitatively supposed. There are several instanes where analysis is restricted to 'eyeballing' relationships between different curves, and correlation coefficients are not presented. In some occasions this leads to the authors making inferences that are not backed up by the data that is presented.*

**Author's response:** We have added tables of Type III effect summaries for all models in the supplementary materials.

**Reviewer's comment:** *The writing is overly descriptive, and often disconnected with the conclusions. Is this study describing entropy fluxes and efficiency ratios and how these vary with different environmental conditions, or is it trying to use these variables to understand site differences? The result is an unclear combination of the two. I would recommend the Authors' to stick to a storyline that is supported by the data.*

**Author's response:** Thank you for this valuable comment. We used environmental variables to understand changes and differences in entropy production and fluxes and thus changes in energy efficiency at our three sites. We have added an explanation of the objective of this study.

**Reviewer's comment:** *Finally, there are several instances where the authors discuss the effect of soil moisture and rainfall on various fluxes/processes in the text (e.g. lines 13,19, 31 on page 10, line 25 on page 11) but do not choose to show these data. In my opinion these data are critical and need to be discussed (since it is a drought recovery study).*

**Author's response:** We have added figures showing all significant effects included in our models.

**Reviewer's comment:** *In light of these observations, I would not recommend this manuscript in its current form for publication in Biogeosciences. I think the authors provide very valuable observations, but should consider either re-framing the study or provide a more critical discussion on the concept of MEP, as well as consider extensive revisions on the writing as well as presentation of data.*

**Author's response:** We have altered the introduction and discussion to reflect your comment.

**Figures**

**Reviewer's comment:** *There are several instances where curves are classified as significantly different, but do not appear significantly different from each other at all (Fig. 1d, for instance). The authors need to expand figure captions, since in the current form it is hard to infer what is being shown. E.g. Figure 4 has three time series (one for each site in most panels) but only one for sub panels b and e. It is unclear what data are presented. There are similar issues with Figs. 5-7.*

**Author's response:** We have added supplementary tables with all type 3 effects for all models included in this manuscript to show where there are significant differences among the independent variables. However, we note in figure captions when interactive effects were not significant, thus only showing a single black line (for example see fig. 6 for panels (g), (h) and (o)).

**Reviewer's comment:** *I also feel that the authors rely on too much on summarizing data and do not explain how or why this is done (again, eg. Fig 4b and d). What are the data that are presented in these analyses?*

**Author's response:** We have changed our analysis to estimating entropy from mean half hourly energy fluxes to daily time-steps (W m$^{-2}$ K$^{-1}$).

**Reviewer's comment:** *The authors need to include sub panels in the text (Fig. 4a, b etc.).*

**Author's response:** Thank you for this suggestion; we have added sub-panel information to the text.

**Reviewer's comment:** *Figure 1 has inconsistent units for temperature. For instance, subpanels c and e are plotted in units of Kelvin but d and f are in deg. C. Also, VPD is plotted in Figure 1 but not discussed at all amongst other discussions of Fig. 1 (Sec. 3.1).*

**Author's response:** We have added Figure 2 for sky, air, surface and soil temperatures with consistent units. We have also added more text describing differences in VPD (see section 3.1).

**Reviewer's comment:** *Fig 2. Why are monthly means shown here, while the rest of the paper annual means are presented?*

**Author's response:** We are now using daily average half hourly estimates in our models. All figures will show these values on the same timestep, with the exception of SI Figure S1, as it seemed more appropriate to show monthly sums of rainfall, as differences among the sites and years became more apparent this way.

**Reviewer's comment:** *Table 1: Please provide LAI estimates (if available) and also disturbance history, since this is a key component of your overall conclusions.*

**Author's response:** We have added LAI data for the mesic and xeric site. Unfortunately that information was not available for the intermediate site. We have also added fire disturbance history for all sites to Table 1.

**Minor comments**

**Reviewer's comment:** *Page 2 Line 1-2: Turbulent exchange of... specify (for e.g. momentum, heat, gases). Line 3: Maybe just use examples related to terrestrial ecosystems? Are these examples of the butterfly effect in terrestrial ecosystems?*

**Author's response:** We have adjusted the sentence accordingly.

**Reviewer's comment:** *Page 5 Lines 5-9: This assumes energy balance closure. Please describe why you closed the energy balance.*

**Author's response:** To accurately describe the entropy balance for ecosystems, we are required to have a closed energy balance. Following your and reviewer 1's comments, we have added a more detailed description of the energy balance and energy partitioning at the three sites (see section 2.2)

**Reviewer's comment:** *Page 6 Eq. 2: Describe briefly how NEE was partitioned into source and sink terms.*

**Author's response:** We have added a more detailed description on how we partitioned NEE into GEE and $R_{eco}$ following Whelan et al. (2013) and Starr et al. (2016).

**Reviewer's comment:** *Page 7 eq. 3.6. and 3.7: Unclear why net fluxes are used. Line 23: Are periods of rainfall excluded from the analyses? Where is this described? eq 4.1 and 4.2: Why is 4.1. formulated using incoming radiation whereas as 4.2 using net fluxes?*

**Author's response:** We have revised this section and are estimating entropy production as follows:

$$\sigma Rs = Rs, in - Rs, out \left( \frac{1}{Tsrf} - \frac{1}{Tsun} \right)$$

$$\sigma Rl = Rs, in \left( \frac{1}{Tsrf} - \frac{1}{Tsun} \right)$$

For the actual entropy production calculation only shortwave radiation that was absorbed by the ecosystem would be converted into heat, whereas for the MEP calculation we quantified the maximum entropy production, assuming that an efficient ecosystem would absorb all $R_{s,in}$. In contrast, for $R_{l,in}$ energy is absorbed and then reemitted by the ecosystem rather than reflected.

**Reviewer's comment:** *Page 8 eq. 4.8 is essentially carbon use efficiency (see major comment above).*

**Author's response:** We have omitted the section on metabolic ratios and focus on energy and entropy of the metabolic system using NEE.

**Reviewer's comment:** *Page 9 Line 11. Subpanels missing. Lines 21-24: temperatures differences do not appear to be significantly different across sites in Fig. 1.*

**Author's response:** We kindly note that even though these figures appear to not show significant differences in temperature across the sites, our statistical results indicate that there were in fact significant differences in temperature. We have added supplementary type-3 results for all models to show this.

**Reviewer's comment:** *Page 10. Sec. 3.2. Methods for this analysis are not presented. I think this section should be merged with Sec. 2.1. (site description), as it doesn't appear to be a result of this study (unless methods are presented). Line 14: Soil moisture data seems important here (and in other places). Line 15: VPD effects are discussed first but EVI figure shown first in Fig. 4. Line 23: This is not correct according to Fig. 4. Line 23: See major comment above.*

**Author's response:** We have added a description of the methods that were used to estimate understory biomass for the three sites. In addition we have added graphs for all significant effects in the models. The description of all variables has been ordered to be consistent with the text.

**Reviewer's comment:** *Page. 13 Line 1: What does 'preservation' on LE mean? Again, these are hard to interpret in the absence of absolute fluxes (see major comment above). Line 8: Ecosystems do not 'experience' LE (or JLE), but rather the interactions between the ecosystem and the overlying atmosphere determines the LE flux. Line 13: Clarify what this means.*

**Author's response:** As noted above, we have added an analysis of the energy balance to show absolute fluxes. We also have changed the wording and made sure to not "personalize" ecosystems throughout this manuscript.

**Reviewer's comment:** *Page 14 Line 8: should read "at the more biodiverse site (i.e. mesic)" Line 11: What was the contribution of the C4 understory photosynthesis to overall ecosystem photosynthesis? Did you measure this? Lines 25-30: This is incorrect. Annual (and monthly) changes in EVI do not reflect changes in biomass. Biomass includes the carbon stored in the trunks, branches and stems of trees (among other pools), which do not fluctuate in forests at these timescales. Instead, at these timescales EVI is a measure of canopy greenness that is related to net photosynthesis (see Sims et al., 2008).*

**Author's response:** We have changed the sentence accordingly. Unfortunately, we have no estimates of variation in ecosystem fluxes from differences in understory composition at our sites. However, in another study we showed that the understory contributes about 50% to $R_{eco}$, using soil respiration data. We have also corrected the definition of EVI in the text.

[revised manuscript text omitted]
 3 effects for models of metabolic energy (NEE$_\varepsilon$) and entropy (J$_{NEE}$)

| Effect | Chisq | | Df | Pr(>Chisq) | Effect |
|---|---|---|---|---|---|
| | Year | 29.646 | 7 | 0.0001102 |
| | Month | 74.127 | 11 | < 0.001 |
| | SWC | 19.826 | 1 | < 0.001 |
| | Site | 779.838 | 2 | < 0.001 |
| | EVI | 75.114 | 1 | < 0.001 |
| | Rain | 300.884 | 1 | < 0.001 |
| **NEE$_\varepsilon$** | VPD | 327.07 | 1 | < 0.001 |
| | Month:Site | 742.229 | 22 | < 0.001 |
| | Site:EVI | 14.519 | 2 | 0.0007 |
| | Site:VPD | 11.067 | 2 | 0.0034 |
| | Site:Rain | 42.48 | 2 | < 0.001 |
| | Year:Site | 520.107 | 14 | < 0.001 |
| | Year | 100.0912 | 7 | < 0.001 |
| | Month | 734.3098 | 11 | < 0.001 |
| | SWC | 102.5001 | 1 | < 0.001 |
| | Site | 472.4768 | 2 | < 0.001 |
| | EVI | 123.4161 | 1 | < 0.001 |
| | Rain | 85.0485 | 1 | < 0.001 |
| **J$_{NEE}$** | VPD | 839.139 | 1 | < 0.001 |
| | Month:Site | 675.38 | 22 | < 0.001 |
| | SWC:Site | 24.5701 | 2 | < 0.001 |
| | Site:VPD | 9.1967 | 2 | 0.0101 |
| | Site:Rain | 22.9547 | 2 | < 0.001 |
| | Year:Site | 3070.236 | 14 | < 0.001 |

**Table S6:** Type 3 effects for models of entropy efficiency

| Model | Effect | Chisq | Df | Pr(>Chisq) |
|---|---|---|---|---|
| $eff_{rad}$ | Site | 321.3179 | 2 | < 0.001 |
| | Year | 28.9597 | 7 | 0.0002 |
| | Month | 31.1969 | 11 | 0.001 |
| | VPD | 269.8436 | 1 | < 0.001 |
| | Rain | 295.7158 | 1 | < 0.001 |
| | SWC | 6.6371 | 1 | 0.001 |
| | Site:Month | 78.91 | 22 | < 0.001 |
| | Site:VPD | 10.683 | 2 | 0.0048 |
| | Site:Rain | 17.7766 | 2 | 0.0001 |
| | Site:Year | 165.2804 | 14 | < 0.001 |
| $eff_{flux}$ | Mite | 938.8639 | 2 | < 0.001 |
| | Year | 9.2791 | 7 | 0.2332 |
| | Month | 251.1215 | 11 | < 0.001 |
| | VPD | 1204.1726 | 1 | < 0.001 |
| | EVI | 5.4535 | 1 | 0.0195 |
| | Rain | 122.5276 | 1 | < 0.001 |
| | SWC | 8.9111 | 1 | 0.0028 |
| | Site:Month | 307.582 | 22 | < 0.001 |
| | Site:SWC | 25.8864 | 2 | < 0.001 |
| | Site:VPD | 17.4305 | 2 | 0.0002 |
| | Site:Rain | 51.4031 | 2 | < 0.001 |
| | Site:EVI | 15.1919 | 2 | 0.0005 |
| | Site:Year | 517.3889 | 14 | < 0.001 |
| dS/dt | Site | 12.945 | 2 | 0.0016 |
| | Year | 5.9043 | 7 | 0.551 |
| | Month | 16.8799 | 11 | 0.1115 |
| | VPD | 114.1762 | 1 | < 0.001 |
| | EVI | 4.207 | 1 | 0.0403 |
| | Site:Month | 103.0141 | 22 | < 0.001 |
| | Site:Year | 135.7525 | 14 | < 0.001 |

[Figure]

**Figure S1:** Monthly rainfall sums and Palmer Drought Severity Index (PDSI) for the mesic, intermediate and xeric sites from 2009 through 2016.

[Figure]

**Figure S2:** Least square mean predicted values from the mixed model results for annual (a) and monthly (b) changes of the energy fluxes of net radiation ($R_n$), latent energy (LE), sensible heat (H), and ground heat (G) at the mesic, intermediate and xeric sites. Error bars represent standard errors (SE).

[Figure]

**Figure S3:** Least square mean predicted values from the mixed model results for interactive effects of (a, e, i, m) enhanced vegetation index (EVI), (b, f, j, n) soil water content (SWC), (c, g, k, o) vapor pressure deficit (VPD) and (d, h, l, p) rain on the energy fluxes of (a-d) net radiation ($R_n$), (e-h) latent energy (LE), (i-l) sensible heat (H), and (m-p) ground heat (G). For (d) and (h) the interaction with site was not significant, as indicated by a single solid black line. Error bars represent SE.

[Figure]

[Figure]

**Figure 7:** Differences in the entropy efficiencies by for radiation entropy efficiency ($eff_{rad}$) in interaction with site and (a) soil water content (SWC), (b) vapor pressure deficit (VPD), and (c) air temperature ($T_{air}$), as well as (d) metabolic entropy efficiency ($eff_M$) in interaction with site and SWC.

| Page 2: [1] Deleted | revised | 12/15/18 1:09:00 PM |
|---|---|---|

| Page 2: [2] Deleted | revised | 12/15/18 1:09:00 PM |
|---|---|---|

| Page 2: [3] Deleted | revised | 12/15/18 1:09:00 PM |
|---|---|---|

| Page 8: [4] Deleted | revised | 12/15/18 1:09:00 PM |
|---|---|---|

| Page 8: [4] Deleted | revised | 12/15/18 1:09:00 PM |
|---|---|---|

| Page 8: [4] Deleted | revised | 12/15/18 1:09:00 PM |
|---|---|---|

| Page 8: [5] Deleted | revised | 12/15/18 1:09:00 PM |
|---|---|---|

| Page 8: [5] Deleted | revised | 12/15/18 1:09:00 PM |
|---|---|---|

| Page 8: [5] Deleted | revised | 12/15/18 1:09:00 PM |
|---|---|---|

| Page 8: [6] Deleted | revised | 12/15/18 1:09:00 PM |
|---|---|---|

| Page 8: [6] Deleted | revised | 12/15/18 1:09:00 PM |
|---|---|---|

| Page 8: [6] Deleted | revised | 12/15/18 1:09:00 PM |
|---|---|---|

| Page 8: [6] Deleted | revised | 12/15/18 1:09:00 PM |
|---|---|---|

| Page 8: [7] Deleted | revised | 12/15/18 1:09:00 PM |
|---|---|---|

| Page 8: [7] Deleted | revised | 12/15/18 1:09:00 PM |
|---|---|---|

| Page 8: [7] Deleted | revised | 12/15/18 1:09:00 PM |
|---|---|---|

| **Page 8: [7] Deleted** | **revised** | **12/15/18 1:09:00 PM** |

| **Page 8: [7] Deleted** | **revised** | **12/15/18 1:09:00 PM** |

| **Page 8: [7] Deleted** | **revised** | **12/15/18 1:09:00 PM** |

| **Page 8: [7] Deleted** | **revised** | **12/15/18 1:09:00 PM** |

| **Page 8: [7] Deleted** | **revised** | **12/15/18 1:09:00 PM** |

| **Page 8: [7] Deleted** | **revised** | **12/15/18 1:09:00 PM** |

| **Page 8: [7] Deleted** | **revised** | **12/15/18 1:09:00 PM** |

| **Page 8: [7] Deleted** | **revised** | **12/15/18 1:09:00 PM** |

| **Page 8: [8] Deleted** | **revised** | **12/15/18 1:09:00 PM** |

| **Page 8: [8] Deleted** | **revised** | **12/15/18 1:09:00 PM** |

| **Page 8: [8] Deleted** | **revised** | **12/15/18 1:09:00 PM** |

| **Page 8: [8] Deleted** | **revised** | **12/15/18 1:09:00 PM** |

| **Page 8: [8] Deleted** | **revised** | **12/15/18 1:09:00 PM** |

| **Page 8: [8] Deleted** | **revised** | **12/15/18 1:09:00 PM** |

| **Page 8: [8] Deleted** | **revised** | **12/15/18 1:09:00 PM** |

| **Page 8: [8] Deleted** | **revised** | **12/15/18 1:09:00 PM** |

| Page 8: [8] Deleted | revised | 12/15/18 1:09:00 PM |
|---|---|---|

| Page 8: [8] Deleted | revised | 12/15/18 1:09:00 PM |
|---|---|---|

| Page 8: [8] Deleted | revised | 12/15/18 1:09:00 PM |
|---|---|---|

| Page 8: [8] Deleted | revised | 12/15/18 1:09:00 PM |
|---|---|---|

| Page 8: [8] Deleted | revised | 12/15/18 1:09:00 PM |
|---|---|---|

| Page 9: [9] Deleted | revised | 12/15/18 1:09:00 PM |
|---|---|---|

| Page 9: [10] Deleted | revised | 12/15/18 1:09:00 PM |
|---|---|---|

| Page 9: [11] Deleted | revised | 12/15/18 1:09:00 PM |
|---|---|---|

| Page 9: [12] Deleted | revised | 12/15/18 1:09:00 PM |
|---|---|---|

| Page 9: [13] Deleted | revised | 12/15/18 1:09:00 PM |
|---|---|---|

| Page 9: [14] Deleted | revised | 12/15/18 1:09:00 PM |
|---|---|---|

| Page 9: [15] Deleted | revised | 12/15/18 1:09:00 PM |
|---|---|---|

| Page 9: [16] Deleted | revised | 12/15/18 1:09:00 PM |
|---|---|---|

| Page 9: [17] Deleted | revised | 12/15/18 1:09:00 PM |
|---|---|---|

| Page 13: [18] Deleted | revised | 12/15/18 1:09:00 PM |
|---|---|---|

| Page 13: [19] Deleted | revised | 12/15/18 1:09:00 PM |
|---|---|---|

| Page 13: [21] Deleted | revised | 12/15/18 1:09:00 PM |

| Page 16: [22] Deleted | revised | 12/15/18 1:09:00 PM |

---

## Author Response (AR2)

College of Arts and Sciences

Department of Biological Sciences

[Figure]

Dr. Gregory Starr, Professor                                    gstarr@ua.edu
Department of Biological Sciences                              205-348-0556
3097A Shelby Hall
Tuscaloosa, AL. 35487

April 1, 2019

Dear Dr. Still

Please find enclosed our second set of revisions for manuscript, mss# bg-2018-322, "Quantifying energy use efficiency via entropy production: A case study from longleaf pine ecosystems." We have completed all revisions per the Reviewer's' comments and suggestion. Below are the Reviewer's' comments in normal black font and our responses to the comments in blue. We have included in our online submission both a marked and clean copy of the revised manuscript. Please contact me if you have any questions regarding the manuscript.

Sincerely,

Gregory Starr, Ph.D.
Professor of Global Change Biology
University of Alabama

1. The entropy production by solar radiation is calculated by Eq. 4.6, and I agree with this formulation. Solar radiation carries the imprints of the solar emission temperature, and is absorbed and produces heat that is associated with the surface temperature, so the expression for entropy production includes the temperatures as it is commonly done. However, later on page 7 the authors use a metric called "Empirical MEP", where instead of the temperature at which solar radiation is being absorbed they use air temperature. I see no physical justification for using air temperature. Solar radiation is absorbed at the surface, not in the air.

Author's Response: Thank you for this comment and you are absolutely correct. We have changed this error in our calculation of MEP for shortwave radiation and are now calculating $\boxed{MEP_{Rs} = R_{S,in}\left(\dfrac{1}{T_{srf}} - \dfrac{1}{T_{sun}}\right)}$. We have updated the results and figures. The results and tests of statistical significance for eff$_{rad}$ did not change.

2. The radiation efficiency they define in eq. 5.4 appears arbitrary to me. In thermodynamics, efficiencies relate to how much meaningful work can be derived out a flux, e.g., for a power plant, how much electricity can be generated out of a heat flux. With entropy fluxes, this actually loses the point. I want to illustrate this with two cases. Imagine case A of a power plant that derives no work out of a flux. Its entropy production by some irreversible processes within the plant is given by the flux and the difference between the inverses of the temperatures at which heat comes in with combustion and goes out of the cooling tower. Case 2 is a power plant that derives the maximum of work out of the flux, which is eventually dissipated into heat. As it turns out, both power plants will actually produce entropy at exactly the same rate, and the only difference is that in case B work is performed, while in case A no work is performed. For such a situation, efficiency has a real meaning, as it allows us to distinguish whether energy is being used to perform work or whether it is wasted. The current paper does not do this kind of analysis, but only focuses on efficiencies based on entropy fluxes. I see no justification for doing so. I do not see a motivation why this should tell us something about why an ecosystem is more or less efficient.

Author's Response: In this study we assumed that a system which would produce more work from incoming energy, would also remain at a lower temperature. In fact, the xeric and intermediate sites had higher surface and air temperatures compared to the mesic site. Similarly, both sites had lower ratios of eff$_{rad}$, thus implying a lower efficiency to convert incoming energy into useful work. More precisely, this may indicate that exergy at these sites was greater, given that all sites receive the same amount of radiation (due to their close proximity to one another). So, for our ecosystems of interest, this relates to radiative entropy and the carbon efficiency. When assuming that an ecosystem would use more incoming energy to produce LE, rather than H, air and soil temperature would be lower, which would maintain a larger gradient to surface temperature. This was further indicated by greater eff$_{rad}$ when SWC and VPD increased, which could be related to greater LE and thus lower surface temperatures. In fact, that is what we see at the mesic site (lower temperatures), suggesting a greater efficiency to convert incoming energy into work; in contrast, the temperature gradient between soil temperature and $T_{srf}$ at the intermediate and xeric sites is much lower, as a consequence of lower efficiency. This may be compared to a solar power system, which becomes more efficient when the solar panels are cooler. Additionally, outgoing shortwave radiation was much greater at the xeric and intermediate sites, indicating lower absorption of solar radiation.

So rather than thinking about this problem in terms of a specific quantity of efficiency, here we are looking at the systems and their thermodynamic environment as a whole.

3. I still disagree with using NEE to infer entropy fluxes. Photosynthesis creates chemical free energy out of sunlight. Hence, the absorption of solar radiation does not quite produce as much entropy as a green plate that has the same albedo as a leaf, because some of the radiation is not turned into heat, but rather ends up in the chemical free energy associated with carbohydrates and oxygen. Only when these carbohydrates are being respired is heat being generated and entropy being produced. When you use the combined flux NEE, you would diagnose negative entropy fluxes during the day, and positive during the night. But no process produces negative entropy. So these fluxes need to be separated to make sense, and the entropy production by solar radiation would need to be adjusted to account for the fraction of radiation that is not converted into heat.

Author's Response: We have adjusted the section on metabolic entropy production using absorbed photosynthetic active radiation (page 7, equations 2.16-2.22). The results and discussion sections and figure 7 were edited accordingly.

There is still a sentence on page 8, L1 that claims that "under ideal conditions, an ecosystem maximizes its entropy production when it converts all incoming Rs and Rl into work". No, all incoming radiation cannot be converted into work as it would violate the second law of thermodynamics. I think I already explained this in my last review.

Author's Response: We have deleted the sentence in the revised document.